# Auto-Regressive vs Flow-Matching: a Comparative Study of Modeling Paradigms for Text-to-Music Generation

**Or Tal**                                                                    *or.tal1@mail.huji.ac.il*
*The Hebrew University*
*Meta Fundamental AI Research*

**Felix Kreuk l**                                                             *felixkreuk@meta.com*
*Meta Fundamental AI Research*

**Yossi Adi**                                                                 *yossi.adi@mail.huji.ac.il*
*The Hebrew University*
*Meta Fundamental AI Research*

**Reviewed on OpenReview:** *https://openreview.net/forum?id=xXc5DeaBYw*

## Abstract

Recent progress in text-to-music generation has enabled models to synthesize high-quality musical segments, full compositions, and even respond to fine-grained control signals, e.g. chord progressions. State-of-the-art (SOTA) systems differ significantly in many dimensions, such as training datasets, modeling paradigms, and architectural choices. This diversity complicates efforts to evaluate models fairly and identify which design choices influence performance the most. While factors like data and architecture are important, in this study we focus exclusively on the modeling paradigm. We conduct a systematic empirical analysis to isolate its effects, offering insights into associated trade-offs and emergent behaviors that can guide future text-to-music generation systems. Specifically, we compare the two arguably most common modeling paradigms: auto-regressive decoding and conditional flow-matching. We conduct a controlled comparison by training all models from scratch using identical datasets, training configurations, and similar backbone architectures. Performance is evaluated across multiple axes, including generation quality, robustness to inference configurations, scalability, adherence to both textual and temporally aligned conditioning, and editing capabilities in the form of audio inpainting. This comparative study sheds light on distinct strengths and limitations of each paradigm, providing actionable insights that can inform future architectural and training decisions in the evolving landscape of text-to-music generation. Audio sampled examples are available at: https://huggingface.co/spaces/ortal1602/ARvsFM

## 1 Introduction

Unlike text and vision domains, the audio domain, and music generation in particular, has not yet converged on a dominant modeling approach. While both Auto-Regressive (AR) and non-AR methods have shown strong results, the trade-offs between them remain under explored (Zhu et al., 2023; Pathariya et al., 2024; Li et al., 2025). In natural language processing, the dominant modeling paradigm is AR generation over discrete token sequences (Touvron et al., 2023; Liu et al., 2024). In computer vision, leading models are typically non-AR, relying on diffusion or flow-matching processes over continuous latent spaces (Podell et al., 2023; Liu et al., 2023). However, it is not clear what approach should we follow for music and audio generation. Copet et al. (2023); Agostinelli et al. (2023) demonstrate impressive performance following the

Table 1: A concise summary of our conclusions: Auto-regressive (AR) vs Flow-Matching (FM).

| Axis | Takeaway |
|---|---|
| Text-to-music fidelity *(Sec. 5.1)* | Both modeling paradigms exhibit comparable performance with a slight favor toward AR. The chosen latent frame rate shows a large impact over performance regardless of the length of the latent sequence. |
| Control adherence *(Sec. 5.2)* | AR follows temporally-aligned conditioning more accurately than FM though it is prone to accumulated errors (mismatch of melody-chords). Both paradigms demonstrate a controllability–fidelity trade-off. |
| Music inpainting *(Sec. 5.3)* | Supervised inpainting FM yields the smoothest and most coherent edits. A text-to-music FM model could be used for zero-shot inpainting but would require a hyper-parameter search per-sample or a better sampling strategy to provide more stable outputs. |
| Inference speed and batch scaling *(Sec. 5.4)* | Considering inference on a single A100 GPU, AR with KV-cache mainly benefits from scaling the batch size to $\geq 64$ for sequence durations $\leq 20$ seconds and degrades for longer sequences due to accumulating overheads. This suggest that AR models would probably be beneficial for systems expecting large demands, e.g. integration of a generative model in social media platforms. FM demonstrated faster inference in all other cases for the observed setup. |
| Sensitivity to training configuration *(Sec. 5.5)* | When the number of update steps is capped at 500k, FM reaches near-topline (Sec. 5.1) quality using batch size $\geq 32$, though its text-match keeps improving with scale. The AR model needs a larger token budget per update step to match its topline performance. Our observations suggest that both modeling paradigms would benefit from large-scale training, but FM could offer a more budget-friendly performance trade-off. |
| Limitations *(Sec. 6)* | This study is centered on a 400M-parameter transformer and maintained a controlled experimental setup across all evaluations. We acknowledge that alternative sampling strategies, training methods, model architecture or scale could yield different results. |

AR approach using discrete audio representation. In contrast, Chen et al. (2024); Lan et al. (2024) follows the diffusion and flow matching approaches and also show impressive performance. Lastly, Li et al. (2024); Bai et al. (2024) proposed hybrid methods utilizing AR and non-AR methods.

While a growing number of systems have demonstrated compelling capabilities in text-conditioned music generation, it is unclear what fundamentally accounts for performance differences across models. Variations in training data, latent representations, architecture design, and optimization procedures often confound evaluation. As a result, there is little consensus on whether improvements arise from the modeling paradigm itself or from external factors, like the training data or architectural choices. These contrasts underscore the need for systematic comparison in audio modeling, where foundational choices are still in flux.

To mitigate that, we present a controlled empirical study comparing the two prominent and commonly used approaches for generative modeling in text-to-music generation: AR and Conditional Flow Matching (FM) (non-AR). All models are trained from scratch using the same training data, latent representations, and similar transformer model backbone architectures. We evaluate each modeling paradigm across multiple axes including perceptual quality (section 5.1), adherence to temporal controls (section 5.2), editing capabilities in the form of audio inpainting (section 5.3), inference efficiency (section 5.4) and robustness to training configuration (section 5.5). This design isolates the modeling approach as the primary experimental variable.

Our results highlight consistent differences between the two paradigms, and aims to derive actionable insights to improve future text-to-music generation systems. Auto-regressive models exhibit slightly higher perceptual quality and demonstrate stronger temporally-aligned control adherence, while flow-matching offers faster inference in most cases and also demonstrate better flexibility for editing tasks. Such findings and additional

observations outlined in this work provide practical guidance for selecting modeling paradigms in future music generation systems and are broadly covered in the following sections. Table 1 draws a summarized overview of our main conclusions.

## 2 Related Work

The idea of generating music through artificial intelligence has evolved significantly, beginning with rule-based symbolic systems (Pinkerton, 1956; Papadopoulos & Wiggins, 1999; Donnelly & Sheppard, 2011) and progressing to deep-learning approaches capable of synthesizing high-fidelity audio (Agostinelli et al., 2023; Huang et al., 2023; Copet et al., 2023; Li et al., 2024; Evans et al., 2025). Early experiments in AI-driven music creation focused on MIDI-based outputs (Huang & Wu, 2016), with systems like Jukedeck[1] offering genre-specific composition based on user-defined prompts. However, these early models struggled to capture the richness of human-composed music due to their reliance on pre-structured symbolic representations.

The emergence of deep learning revolutionized this field, first introducing deep Recurrent Neural Network systems Simon & Oore (2017); Mao et al. (2018), which further evolved to transformer-based architectures, such as MuseNet [2] and MusicTransformer (Huang & Guo, 2019), demonstrating that transformers could generate stylistically coherent compositions. Subsequent works that followed moved beyond MIDI-based approaches to generate raw audio waveforms mainly do so using one of three generative paradigms: AR decoding, diffusion, and flow-matching.

A major breakthrough was introduced by JukeBox(Dhariwal et al., 2020), which incorporated both instrumental and vocal elements using AR decoding. Following this line of work, Borsos et al. (2023) introduce AudioLM, which first compresses audio into "semantic" and "acoustic" tokens and uses an AR Transformer to predict them. This enables the model to extend a musical excerpt without relying on any symbolic representation. Building on this idea, MusicLM (Agostinelli et al., 2023) adds a text-to-semantic stage and a hierarchical AR decoder, generating 24kHz music with noticeably better coherence and fidelity than Juke-Box. MusicGen (Copet et al., 2023) simplifies the pipeline, encoding a 32kHz audio into EnCodec (Défossez et al., 2022) tokens and trains a single-stage AR Transformer to predict all token streams jointly, reducing latency while maintaining high prompt adherence and audio quality.

In parallel, inspired by recent success in text-to-image synthesis, diffusion models have emerged as an alternative paradigm, generating music by iteratively refining noise through a learned denoising process. Noise2Music (Huang et al., 2023) introduces a cascaded architecture that enables high-fidelity synthesis through progressive upsampling. MusicLDM (Chen et al., 2024), extends this approach by incorporating beat-synchronous augmentation, improving musical structure alignment with text prompts. StableAudio (Evans et al., 2025) further refines diffusion-based synthesis by introducing low-latency inference and high-resolution output (44.1kHz) offering long generation of full-length songs.

A more recent development is the adoption of flow-based generative models, which learn a continuous transformation from a simple distribution to the target audio distribution while conditioning on text. AudioBox (Vyas et al., 2023) introduces a FM framework capable of handling multiple audio modalities, including music, speech, and environmental sounds. JASCO (Tal et al., 2024) refines FM techniques for music generation, conditioning on textual descriptions and symbolic music features to enhance both coherence and controllability. MelodyFlow (Lan et al., 2024) optimizes single-stage FM models for high-fidelity text-guided music generation, improving both efficiency and musical structure adherence.

An additional timely development is the appearance of approaches combining AR with non-AR (Lam et al., 2023; Li et al., 2024; Bai et al., 2024), yet this falls outside of the scope of this work and will not be considered. Together, these approaches represent a diverse and rapidly evolving landscape in text-to-music generation. AR models continue to set strong baselines for musical structure and coherence (Hawthorne et al., 2022; Huang et al., 2018), while non-AR methods, including Diffusion and FM models, offer promising alternatives for efficient generation and flexible control. Understanding how these paradigms compare under matched conditions remains an open challenge, motivating further exploration.

---

[1] https://techcrunch.com/2015/12/07/jukedeck
[2] https://openai.com/index/musenet/

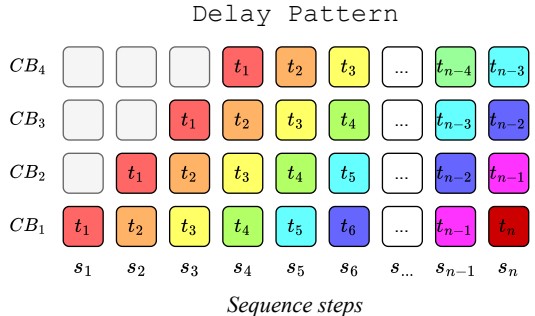

Figure 1: Multi-stream delay pattern modeling. Each row represents a single codebook ($\text{CB}_j$) stream, illustrating the applied delay pattern apperent in the shifting of $\text{CB}_j$ by $j - 1$ sequence steps.

# 3 Background

## 3.1 Problem Setup and Formulation

Given an audio waveform $\boldsymbol{x} \in \mathbb{R}^{f_s \cdot t}$ of duration $t$ seconds, sampled at $f_s$ Hz, we assume access to a pre-trained latent representation model that encodes $\boldsymbol{x}$ into either (i) a continuous latent representation $\boldsymbol{z} \in \mathbb{R}^{D \times f_r \cdot t}$ or (ii) a discrete representation $\boldsymbol{q} \in \mathcal{S}^{N_q \times f_r \cdot t}$ (also known as audio tokenization or audio codec). Here, $f_r$ is the latent frame rate, $D$ the latent dimension, $\mathcal{S}$ is the set of discrete code indices and $N_q$ is the number of parallel code streams. Each discrete code stream has its own embedding table, also referred to as codebook. Our EnCodec model encodes 32kHz music to 50Hz multi-token stream composed of 4 codebook streams. The goal is to train a generative model that operates in this latent space, conditioned on a target textual description, using AR for discrete modeling or FM for continuous modeling. The generation could also be conditioned on other temporally aligned controls in addition to the textual description, e.g. chord progressions.

## 3.2 Auto-Regressive (AR) Decoding

The AR approach models the discrete latent sequence distribution using a causal transformer trained to predict the next token given past context. Given an audio segment $\boldsymbol{x}$, its textual description $c_{\text{txt}}$, and a discrete latent representation of $\boldsymbol{x}$, $\boldsymbol{q} \in \mathcal{S}^{N_q \times f_r \cdot t}$, the model is trained to iteratively generate discrete tokens in an auto-regressive manner. Our discrete latent representation encodes the input waveform $\boldsymbol{x}$ to a sequence of $N_q > 1$ discrete streams that are obtained by utilizing Residual Vector Quantization (RVQ) (Zeghidour et al., 2021; Défossez et al., 2022). RVQ quantizes a continuous encoded latent $\boldsymbol{z} \in \mathbb{R}^{D \times f_r \cdot t}$ to $N_q$ streams of discrete tokens recursively quantizing the current residual. Formally, let $\hat{\boldsymbol{z}}_1$ be the quantized continuous latent representation of $\boldsymbol{z}$ for which each temporal entry was replaced with its closest, in terms of euclidean distance, vector in the $1^{\text{st}}$ codebook and let $\boldsymbol{q}_1$ be the corresponding indices of that sequence. Then, recursively applying quantization we can define $\forall j \in \{2, ..., N_q\} : \hat{\boldsymbol{z}}_j = \boldsymbol{z} - \sum_{l < j} \hat{\boldsymbol{z}}_l$ and obtain the corresponding indices stream $\boldsymbol{q}_j$. This iterative process then yields the discrete multi-stream representation $\boldsymbol{q} = \text{stack}([\boldsymbol{q}_1, ..., \boldsymbol{q}_{N_q}])$.

**Training with Delay Pattern.** Notice, at each timestep, one needs to predict $N_q$ different codes corresponding to different codebooks. This begs the question, *how should we predict these codebooks?*. Naturally, this multi-stream representation dictates an inherent dependence between stream $j$ and all the ones proceeding it. Due to that, predicting $N_q$ codebooks in parallel at each timestep means independently sampling $N_q$ tokens, ignoring the inherently dependent structure. To mitigate that, we follow MusicGen (Copet et al., 2023) and train the model using a *delay pattern* to structure the multi-stream discrete representation. Instead of predicting all codebooks simultaneously, each stream is shifted by a predefined delay, enforcing a temporal offset that allows early predictions to condition later ones.

For a quantized sequence $\boldsymbol{q} \in \mathcal{S}^{N_q \times f_r \cdot t}$ with $N_q$ codebook streams, we impose a delay pattern structure on the multi-stream sequence to allow the $i^{\text{th}}$ codebook at timestep $j$ to be conditioned on all previous $< i$ codebook streams for all $\leq j$ timesteps, as depicted in Figure 1. Formally, denote $n = f_r \cdot t$, then

$\forall i \in [N_q], j \in [n]$, we define a mapping $P : [N_q] \times [n] \to [N_q] \times [n + N_q - 1]$ s.t $P(i, j) = (i, j + i - 1)$. The model outputs a distribution over discrete tokens for each codebook stream across the temporal axis $\boldsymbol{p} \in \mathbb{R}^{N_q \times |\mathcal{S}| \times f_r \cdot t}$, and trained to minimize the cross-entropy objective:

$$\mathcal{L}_{\mathrm{CE}}(\boldsymbol{q}, \boldsymbol{p}) = -\frac{1}{N_q \cdot |\mathcal{S}| \cdot f_r \cdot t} \sum_{k \in [N_q]} \sum_{j \in \mathcal{S}} \sum_{i \in [f_r \cdot t]} \mathbf{1}_{\{\mathrm{q_{k,i}=j}\}} \cdot \log \boldsymbol{p}_{k,j,i}. \tag{1}$$

**Inference Process.** At inference time, the model generates tokens sequentially, following the same delay pattern used in training. Generation starts with an empty context, progressively sampling tokens while respecting the predefined temporal shifts between codebooks. Top-$k$ or top-$p$ sampling strategies are used during decoding, which are standard methods to control diversity in AR generation. The process continues iteratively until the full latent sequence is generated, after which it is decoded back into a waveform using the pretrained audio tokenizer.

### 3.3 Conditional Flow Matching (FM)

FM is an approach to training continuous normalizing flows by regressing a neural network onto a known vector field that generates a probability path (Lipman et al., 2022). FM is optimized to predict a deterministic vector field to directly transform noise (or any other input distribution) to the data distribution.

**Probability Path Definition.** We define a continuous transformation from a simple prior $p_0$ (e.g., $\mathcal{N}(0, I)$) to the data distribution $p_1$ using a time-dependent probability flow $\psi_\tau(y|y_1)$. The probability flow $\psi_\tau(y|y_1)$ defines a continuous path from $y_1$ to $y$ for which $\psi_\tau(y|y_1) = y \sim p_0$ at $\tau = 0$ and $\psi_\tau(y|y_1) = y_1 \sim p_1$ at $\tau = 1$.

Following the optimal transport setup, as defined by Lipman et al. (2022), we define a probability flow from $p_0$ to $p_1$ such that: (i) the mean and variance evolve linearly with time $\tau \in [0, 1]$; (ii) for $y_0 \sim p_0$ and $y_1 \sim p_1$, the probability path $\psi_\tau(y|y_1)$ and its corresponding vector field $\boldsymbol{v}_\tau(y|y_1)$ are given by:

$$\begin{aligned} \psi_\tau(y|y_1) &= (1 - (1 - \sigma_{\min})\tau)y + \tau y_1, \\ \boldsymbol{v}_\tau(y|y_1) &= \frac{y_1 - (1 - \sigma_{\min})y}{1 - (1 - \sigma_{\min})\tau}, \end{aligned} \tag{2}$$

where $\sigma_{\min}$ is some small constant to ensure numerical stability.

**Training Objective.** Rather than modeling the full marginal probability path, the model estimates the conditional vector field $\hat{\boldsymbol{v}}_\tau(y|y_1)$. Given a text-conditioned latent $y_1$ drawn from $p_1$, we train the model to minimize the mean squared error between the estimated and reference vector fields

$$\mathcal{L}_{\mathrm{FM}}(\hat{\boldsymbol{v}}_\tau(y|y_1), \boldsymbol{v}_\tau(y|y_1)) = \mathbb{E}_{\tau \sim [0,1]} \left[ ||\hat{\boldsymbol{v}}_\tau(y|y_1) - \boldsymbol{v}_\tau(y|y_1)||^2 \right]. \tag{3}$$

Following Tal et al. (2024) we slightly modify this training objective by applying a $\tau$ dependent loss scaling, assuming a batch size of $B$ samples:

$$\mathcal{L}_{\mathrm{FM}}(\hat{\boldsymbol{v}}_\tau(y|y_1), \boldsymbol{v}_\tau(y|y_1)) = \frac{1}{B} \sum_{i \in [B]} (1 + \tau_i) \cdot ||\hat{\boldsymbol{v}}_{\tau_i}(y|y_1) - \boldsymbol{v}_{\tau_i}(y|y_1)||^2. \tag{4}$$

**Inference Process.** Generation follows a non-AR iterative process, iteratively refining the vector field estimation $\hat{\boldsymbol{v}}(y_\tau|y_1)$ using an ODE solver. Given an initial sample $y_0 \sim p_0$, we update the state iteratively:

$$y_\tau = y_{(\tau - \Delta\tau)} + \Delta\tau \cdot \hat{\boldsymbol{v}}(y_\tau|y_1). \tag{5}$$

While various ODE solvers and sampling schedules could be applied during inference, in this study we only consider two representative sampling methods: Euler's method for a fixed-grid sampling and Dopri5 (Dormand & Prince, 1980) for a dynamic (adaptive-step) sampling.

**Similarity to Diffusion Modeling.** Diffusion models (Song et al., 2020) and Flow Matching (FM)(Lipman et al., 2022) are both continuous-time generative modeling frameworks that transform a simple initial distribution (usually Gaussian noise) into a target data distribution. Despite originating from different theoretical ideas, they end up with similar training procedures and generation processes. This similarity means results and "actionable insights" from one paradigm (e.g. improved samplers, better weighting of losses, network architecture tweaks) may directly inform the other (Lipman et al., 2024; Gao et al., 2024). For an in-depth derivation of the cases in which this claim applies, refer to Sec. 10 in Lipman et al. (2024).

## 4 Experimental Setup

Throughout this paper we consider text-to-music generation tasks. We also evaluate AR and FM considering temporally aligned conditioning for music generation and music inpainting. As we perform multiple experiments in this work, this section serves to define the common ground. Deviations from the shared experimental setup defined in this section would be explicitly described in each of the relevant subsections.

### 4.1 Data

We train our models on a private proprietary dataset containing tracks from Shutterstock [3] and Pond5 [4] data collections, which sums to roughly 20k hours of Mono 32kHz mixtures paired with textual descriptions. We evaluate the trained models on a different private proprietary data containing 162 hours of high-quality Mono 32kHz mixtures paired with high quality captioning textual descriptions. As we wish to observe subtle changes in performance throughout this work, we refrained from evaluating the models over Music-Caps (Agostinelli et al., 2023) as it is unclear what subtle differences in performance stand for w.r.t this set due to inconsistencies in audio-quality, sampling rates and audio-text match. For further comparison of the data sources used in this work please see Appendix A.

### 4.2 Models

**Input Representation.** To isolate the modeling paradigm, we keep the representation space fixed. En-Codec's (Défossez et al., 2022) latent provides this bridge as the same encoder outputs: (i) discrete indices for AR decoding and (ii) continuous vectors for FM. This design choice follows prior cross-paradigm work, e.g. Copet et al. (2023); Tal et al. (2024). For completeness, we replicate the entire pipeline using StableAudio's (Evans et al., 2025) open source training code[5] considering matched model sizes and frame-rates thereby checking whether FM benefits from an encoder trained without quantization. Note, the configurations of both models were adapted to match the desired frame rates and to have comparable model sizes, hence sub-optimal performance of these is possible. The reconstruction quality comparison of the representation models in Appendix B demonstrates comparable reconstruction quality for both representation models.

**Backbone Model.** For the backbone transformer architecture we use the open source implementation of MusicGen Copet et al. (2023) using a 400M parameters transformer configuration (specifically - 'musicgen-small' [6]). For the FM case we include U-Net-like skip connections, as it is a relatively standard practice in recent years for Diffusion/FM based approaches that doesn't use the Diffusion transformer (DiT) architecture, e.g. Le et al. (2023); Tal et al. (2024); Zhang et al. (2025). The inclusion of such skip connections had a notable impact on performance during preliminary experimentation, and therefore we chose to follow this standard practice and add this slight change to the backbone model, only introducing a small number of additional parameters ($\sim 7M$). We use T5 (Raffel et al., 2020) to obtain text embeddings and pass them via cross-attention layers as text conditions in both cases. For further details regarding the latent representation models and the backbone transformer architecture refer to Appendix E.

---

[3] https://shutterstock.com/music
[4] https://pond5.com
[5] https://github.com/Stability-AI/stable-audio-tools/tree/main
[6] https://github.com/facebookresearch/audiocraft/blob/main/config/model/lm/model_scale/small.yaml

### 4.3 Evaluation Metrics

**Perceptual Quality.** We employ Fréchet Audio Distance (Kilgour et al., 2018) (FAD) with the reference being a high-quality curated proprietary test set. We report FAD using the open source implementation *fadtk*[7] (vggish)(Hershey et al., 2017) where a lower FAD score is associated with a higher perceptual quality.

**Audio Aesthetics.** We use Audiobox Aesthetics (Tjandra et al., 2025) estimators which serves as a proxy for subjective evaluation, evaluating different properties of the generated audio. Specifically, we consider:

- *Production Quality (PQ):* assesses the technical fidelity of the audio, the absence of distortions or other artifacts, well-balanced frequency range and smooth dynamics. It also reflects the skill in recording, mixing, and mastering.

- *Production Complexity (PC):* measures the number of audio elements and their interaction. Higher scores indicate layered compositions with multiple sounds, while lower scores reflect simpler, single-source recordings. It also considers how well elements blend together.

- *Content Enjoyment (CE):* captures the subjective appeal of audio, considering emotional impact, artistic skill, and creativity. Higher scores reflect engaging, expressive, and aesthetically pleasing content.

**Text Description Match.** To evaluate how well the generated audio matches the given textual description we compute the cosine similarity over a joint CLAP (Chen et al., 2022) text-audio representation. The similarity is computed between the track description and the generated audio, measuring audio-text alignment. We use the official pretrained CLAP model (HTSAT-base)[8] in our evaluation.

**Temporally-Aligned Control Adherence.** We evaluate the adherence to different conditions: (i) chord progressions; (ii) melody; and (iii) drum beat conditioning. We extract pseudo-annotations for these conditions using pretrained classifiers and representation models. Detailed description of how these annotations were obtained can be found in Appendix C. We train a model for both AR and FM using all conditions together with condition-dropout.

We consider the following evaluation metrics:

- *Chords Intersection Over Union (IOU)*: We extract chord label and chord switch time in seconds sequence pairs for both the generated and reference audio waveforms and compute the IOU score between the two.

- *Down-beat F1 score (Beat F1)*: As Wu et al. (2024) suggested, we evaluate the down-beat F1 score using *mir eval*[9] (Raffel et al., 2014) considering a 50ms tolerance margin around classified downbeats in the reference signal.

- *Melody Chromagram Cosine Similarity (Melody Similarity)*: Similarly to Copet et al. (2023) and using the officially released implementation, we convert the resulting audio to a 12-bins chromagram representation (single octave) and compute the cosine similarity between the reference and the corresponding chromagrams.

### 4.4 Model Training

Unless stated otherwise, we train all models on 10 second segments for 500k update steps using a batch size of 256, AdamW optimizer with a learning rate of $1 \cdot 10^{-4}$, and a cosine learning rate scheduler with 4000 steps warmup followed by a cosine learning rate decay. We follow the training objectives as defined in Section 3.

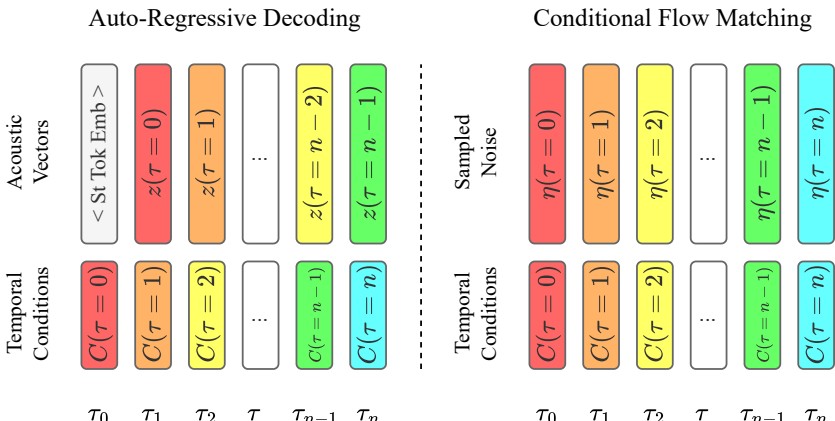

Figure 2: Temporal Conditioning Injection. $\tau_i$ denotes the temporal index in the sequence. In the auto-regressive case we apply a delayed concatenation where the conditions are stacked and concatenated over the channel axis one timestep prior to timestep they correspond to.

## 4.5 Temporally Aligned Conditioning

Following the official release of JASCO (Tal et al., 2024), we explore conditioning using temporally aligned controls, i.e. time-dependent conditions. Controls are resampled to the latent frame-rate $f_r$ and concatenated to the input signal over the channel axis, then passed through a linear layer prior to the transformer module.

As depicted in Figure 2 the conditioning injection is done by concatenating the condition vectors (notated as $C$) to the expected transformer input, which is a shifted input in the AR case (with start token added as the first token) and a sampled standard gaussian noise $\eta$ in the FM case. The concatenated signals are then projected to the expected transformer dimension using a simple linear projection, and fed as input to the transformer module.

The training objective targets for both modeling paradigms remains the same as the text only variants, i.e. AR learns next-token prediction, FM learns the vector field from $\eta$ to $z$. Training is done using a learned new null token for dropout, and inference is done using standard classifier free guidance with unconditional and conditional states.

## 5 Comparative study: AR vs FM

### 5.1 Objective Comparison Under a Fixed Training Setup

We first examine AR and FM modeling paradigms under identical conditions. Each model trains for one million updates on the same dataset and similar backbone model, and we report objective scores at latent frame rates of 25, 50 and 100 Hz in Table 2. Across the three frame rates, both modeling paradigms exhibit comparable performance, with a slight edge toward AR in terms of FAD, PC, and CE, while CLAP and PQ scores remain similar between the two. AR show less fluctuations as latent frame rate increases, overall and for FAD, PQ and CE in particular, compared to FM, especially in the VAE-based latent representation case. Interestingly, within the flow matching models, the EnCodec-based model consistently outperform VAE-based one on every metric. Note, while this observation shows a notable gap in performance, concluding that the EnCodec based space is better for generative modeling is not a direct conclusion. Within the reasonable exploration of training configurations done in this work, the quantized space (EnCodec based) proved to be better suited for modeling, and it serves to diminish the degrees of freedom in our experiments by using the same representation space for both modeling paradigms. As mentioned in Section 4.1 - to

---

[7]https://github.com/microsoft/fadtk

[8]https://github.com/LAION-AI/CLAP

[9]https://github.com/mir-evaluation/mir_eval

Table 2: Objective metrics for autoregressive (AR) and flow matching (FM) models after one million updates. Lower is better for FAD; higher is better for the other metrics.

| Hz | Modeling | FAD↓ | Clap↑ | PQ↑ | PC↑ | CE↑ |
|---|---|---|---|---|---|---|
| | AR | 0.40 | 0.41 | 7.71 | 6.02 | 7.36 |
| 25 | FM (EnC) | 0.42 | 0.39 | 7.78 | 5.42 | 7.13 |
| | FM (VAE) | 0.54 | 0.40 | 7.68 | 5.87 | 7.28 |
| | AR | 0.47 | 0.40 | 7.69 | 5.78 | 7.24 |
| 50 | FM (EnC) | 0.48 | 0.40 | 7.73 | 5.60 | 7.20 |
| | FM (VAE) | 0.87 | 0.40 | 7.41 | 5.87 | 7.07 |
| | AR | 0.64 | 0.40 | 7.59 | 5.84 | 7.17 |
| 100 | FM (EnC) | 0.68 | 0.38 | 7.47 | 5.63 | 6.92 |
| | FM (VAE) | 1.02 | 0.37 | 7.37 | 5.89 | 7.10 |

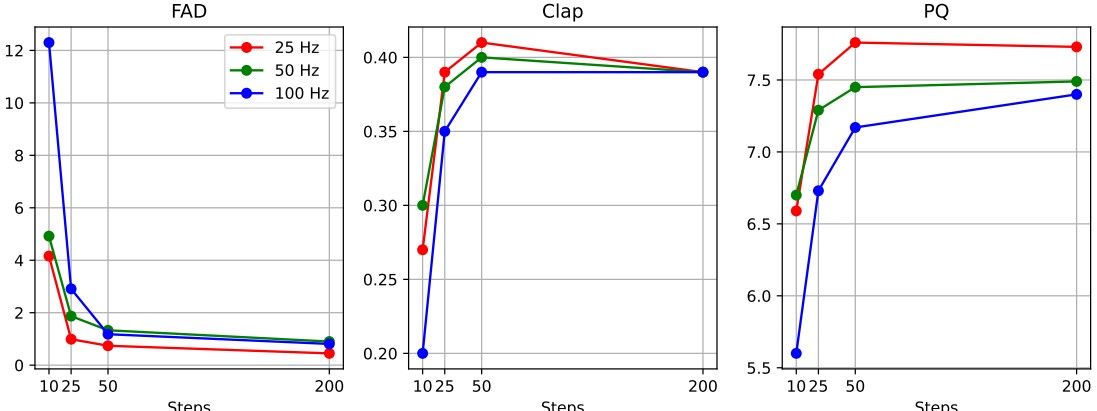

Figure 3: FM performance as a function of inference steps using Euler's method. Decreasing the number of inference steps show a steep degradation in score.

better observe subtle differences in performance - we perform the observed evaluations on a proprietary high-quality evaluation dataset (see Appendix A for further details). Appendix F contains the results of this experiment evaluated on the MusicCaps dataset (Agostinelli et al., 2023), demonstrating a noisier and less concrete comparison that mainly serves to set a reference point for the trained model's performance.

Moreover, both modeling paradigms demonstrate changes in metrics in correspondence with latent frame rate changes, where FM shows slightly more apparent fluctuations. Such differences could stem from two main factors (i) differences in sequence length (ii) the actual latent representation. To better understand the nature of this observation we repeat the experiment for $\{25, 50\}$Hz FM (EnCodec) models considering $\{20, 40\}$sec durations. Under this setup we could isolate the impact of sequence length or the impact of latent frame rate itself; Figure 4 shows the impact over FAD, and the full results table could be found in Appendix G. When fixing the latent frame rate, it appears that extending the sequence length has a consistent impact on performance, although the impact is more apparent at 50Hz than at 25Hz. On the other hand, considering the following pairings: $\{(25\text{Hz}, 20\text{s}), (50\text{Hz}, 10\text{s})\}$, $\{(25\text{Hz}, 40\text{s}), (50\text{Hz}, 20\text{s}), (100\text{Hz}, 10\text{s})\}$ - we fix the length of the sequence the model is required to model; showing a notable gap w.r.t change in representation frame rates. Both of these observations imply that longer sequences could have an impact on performance, though the representation itself serves as the more crucial factor of the two. We hypothesize that such implications

Figure 4: Evaluation results of FM (EnCodec) across varying frame rates (FR) and durations (Dur) - isolating the impact of sequence length and frame rate sensitivity. Full evaluation table is available in Appendix G. Top: fixing latent frame rate and extending the sequence length. Bottom: Fixing sequence length, changing latent frame rate.

stem from the nature of the more compressed representation, where less vectors correspond to the same temporal span - resulting in fewer local dependencies.

The best performing FM configurations follows the Dopri5 (Dormand & Prince, 1980) ODE solver, a dynamic solver that iteratively estimates an error approximation and stops taking additional inference steps if the estimated error is less than a predefined tolerance factor. To further observe how performance is impacted by fixing the number of inference steps we repeat the evaluation using Euler's method on $\{10, 25, 50, 200\}$ inference steps, depicted in Figure 3. As the number of steps reduces below 50 we see a significant degradation in performance. Taking more steps or using the adaptive Dopri5 solver limits this performance degradation, yet still needs a large number of steps to close the performance gap with AR. A full evaluation table is available in Table 8 on the Appendix.

**Take-away.** Both modeling paradigms (EnCodec-based latent) show comparable performance with a slight favor toward AR. The chosen latent frame rate shows a large impact over performance regardless of the length of the latent sequence; an observation that should be taken under consideration when designing a text-to-music generative pipeline. Last, a tradeoff between the number of inference steps and generation quality exists in the FM case; requiring a large number of inference steps to maintain comparable performance.

Table 3: Adherence to temporally aligned controls. "Single" rows report FAD and CLAP for chords/drums/melody conditioning, respectively.

| Modeling | Conditioning | FAD↓ | Clap↑ | Chords IOU↑ | Beat F1↑ | Melody Sim.↑ |
|---|---|---|---|---|---|---|
| AR | All | **0.72** | **0.37** | **0.57** | 0.39 | **0.41** |
| FM | | 0.78 | 0.35 | 0.33 | 0.42 | 0.32 |
| AR | Single | 1.01 / 1.41 / 1.53 | 0.40 / 0.33 / 0.38 | **0.70** | 0.38 | **0.38** |
| FM | | 1.41 / 1.16 / 1.45 | 0.38 / 0.33 / 0.37 | 0.40 | 0.40 | 0.31 |

## 5.2 Temporally Aligned Control Adherence

Next, we compare AR and FM considering temporally aligned conditioning. Following JASCO (Tal et al., 2024), we train 50 Hz models conditioned on three temporally aligned conditions: chord progression, melody, and drum signals. As explained in Subsection 4.5 and visualized in Figure 2, the injection of the temporally aligned conditions is done by concatenating them over the channel axis prior to the transformer module. For each paradigm we test two scenarios: (i) all three controls provided; and (ii) one control provided while the others are set to a null token.

Despite lacking future context, Table 3 shows that the causal AR decoder tracks the controls more faithfully than FM. With all streams active, AR achieves higher Chord IoU (0.57 vs. 0.33) and melody similarity (0.41 vs. 0.32), while Beat F1 is comparable. The pattern holds in the single-control setting: AR leads on chords (0.70 vs. 0.40) and melody (0.38 vs. 0.31), and is on par for drums.

Interestingly, using temporally aligned conditioning reduces overall fidelity as apparent in FAD and CLAP scores in comparison to the text-to-music model in Section 5.1. Relative to the text-to-music model, FAD rises by 0.3–0.8 and CLAP falls by 0.02–0.05 for both paradigms. We hypothesise that the controls act as a strong bias: once top-$p$ sampling ventures onto a low-probability path that still satisfies the controls, the model continues down that trajectory, hurting realism. Appendix I expands this observation further.

It appears that the controllability-fidelity tradeoff is enhanced when two of the three controls are dropped, showing a significant increase in chords IOU. Inherently, melody and chords share information (e.g. the musical key) hence such observation is fairly surprising as the controls are given as conditions. One reason the phenomena could stem from is similar to the observation made for quality - a note that doesn't match the chord was generated (audio of that note), be it due to pseudo-labeling error or sampling process, and from that point the local environment is being pushed toward a different chord. The same apply for sampling of a "wrong" chord, or shifts in rhythm. We believe that this observation stems from suboptimal conditioning in this case, where the AR modeling proves to be more prone to such contradicting controls yet notably adheres better to melody and chords controls.

**Take-away.** AR follows temporally-aligned conditioning more accurately than FM, though it appears to be more prone to accumulated errors (mismatch of melody-chords). Both paradigms lose perceptual quality under strict controls, illustrating a controllability–fidelity trade-off.

## 5.3 Inpainting

Music editing often requires replacing a flawed passage while preserving the surrounding context. We therefore compare inpainting capabilities: generating a masked span given past and future audio context. FM supports naïve zero-shot (ZS) inpainting via latent inversion, whereas AR does not (at least not for the observed vanilla setup). To enable AR we adopt the *fill-in-the-middle* strategy of Bavarian et al. (2022), where special tokens split each training example into $A\,|\,B\,|\,C$ segments; we present the model with $A\,|\,C$ and ask it to generate $B$ causally. For a fair comparison we train both AR and FM. We use a fixed 5 second masked span whose start time is chosen uniformly with at least 1 second margin on both sides. The algorithms used for FM inpainting (supervised and ZS) are available in Appendix J.

Table 4: Objective scores for inpainting with a 5 seconds mask. Lower is better for FAD; higher is better for all other metrics.

| Model | FAD↓ | CLAP↑ | PQ↑ | PC↑ | CE↑ |
|---|---|---|---|---|---|
| AR | **0.23** | 0.36 | 7.75 | 5.65 | 7.31 |
| FM | 0.32 | 0.36 | 7.80 | 5.48 | 7.31 |
| FM (ZS) | 0.30 | **0.39** | **7.89** | **5.73** | **7.40** |

Table 5: Human ratings ($1-10$; mean $\pm$ $95\%$ confidence interval, $\sim 250$ judgments per cell).

| Criterion | GT | AR | FM | FM (ZS) |
|---|---|---|---|---|
| Transition smoothness | 8.78±0.10 | 7.57±0.19 | **8.11±0.15** | 7.09±0.26 |
| Audio match | 8.81±0.12 | 7.22±0.29 | **7.93±0.21** | 6.78±0.37 |

Table 4 shows that all three approaches achieve similar objective scores. However, while listening to the generated audio, we notice artifacts, audible glitches or mismatched timbre, that the objective metrics fail to capture. To support our subjective observations, we conducted a human study, in which the raters were requested to rank each of the observed methods on a scale of 1 to 10, where higher is better. Each audio segment was accompanied with its corresponding waveform figure and a horizontal red line indicator in correspondence to the current temporal position. The inpainted segment was visibly marked with a **yellow** background. For a visualization example see our Sample page. The raters were required to evaluate the following criteria:

- **Transition smoothness**: How smooth the transitions between the yellow (inpainted) and the white (reference) segments is? please refer only to the transitions between the segments.

- **Audio match**: How well does the audio in the yellow segments match the audio in the white segments? A good match should maintain instrumentation, dynamics (volume), tempo, and feel like a part of the same musical piece. Ignore the smoothness of the transition between segments.

The results of the human study, presented in Table 5, confirms preliminary observation. Supervised FM receives the highest scores for both *transition smoothness* and *audio match*, indicating that it generates missing segment with better alignment to the context. AR ranks second: it produces segments with good fidelity (lowest FAD) but often leaves a discernible seam at the boundaries. Zero-shot FM delivers the best CLAP, PQ, and CE but exhibits high variance: some samples perfectly fits the context while others drift into unrelated content. This suggests that the sampling configuration could be updated per-sample and a more complex sampling strategy could be used to improve ZS capabilities.

**Take-away.** Supervised inpainting FM is the best method among the observed approaches yielding the smoothest and most coherent edits. Text-to-music FM could be used for zero-shot inpainting but would require a hyper-parameter search per-sample or a better sampling strategy to provide more stable outputs.

### 5.4 Runtime Analysis and Model Scaling

In subsection 5.1 we show that FM can closely match AR quality when it runs a large number of inference steps and still lags slightly in FAD, PC, and CE. This raises two practical questions: (i) *Is it worthwhile to cut the step count to gain inference speed?*; (ii) *Does such speed-up scales to batch processing?* AR re-uses hidden states through key–value (KV) caching, so its cost per token falls as the batch grows; FM has no comparable mechanism.

To answer these questions, we record throughput (samples / sec) and per-sample latency on a single A100 GPU for batch sizes 2–256; in this context "sample" refers to a generated 10 sec audio sample. AR is

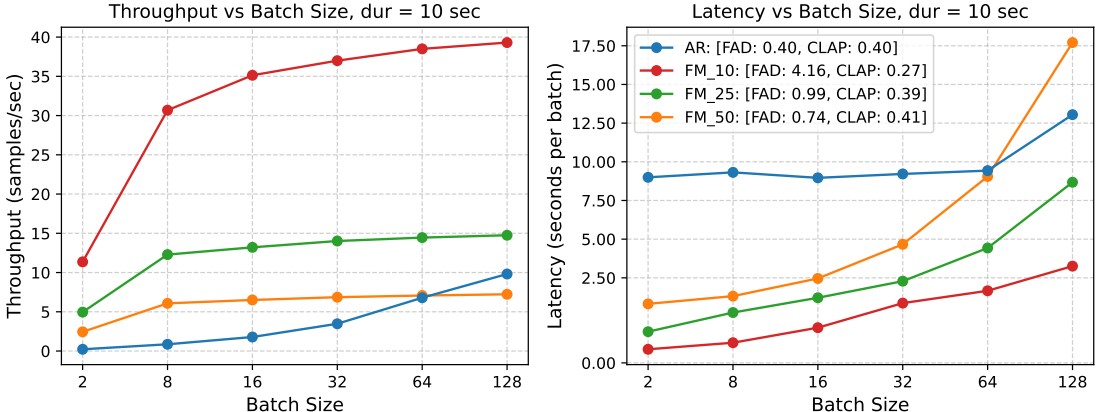

Figure 5: Inference speed versus batch size for 10 sec segments. Left: throughput; right: latency. The "Sample" unit refers to a complete generated 10 sec example. AR gains steadily from KV caching, whereas FM plateaus after batch size 8. Euler's method using 10 steps is the fastest but has the worst FAD (4.16).

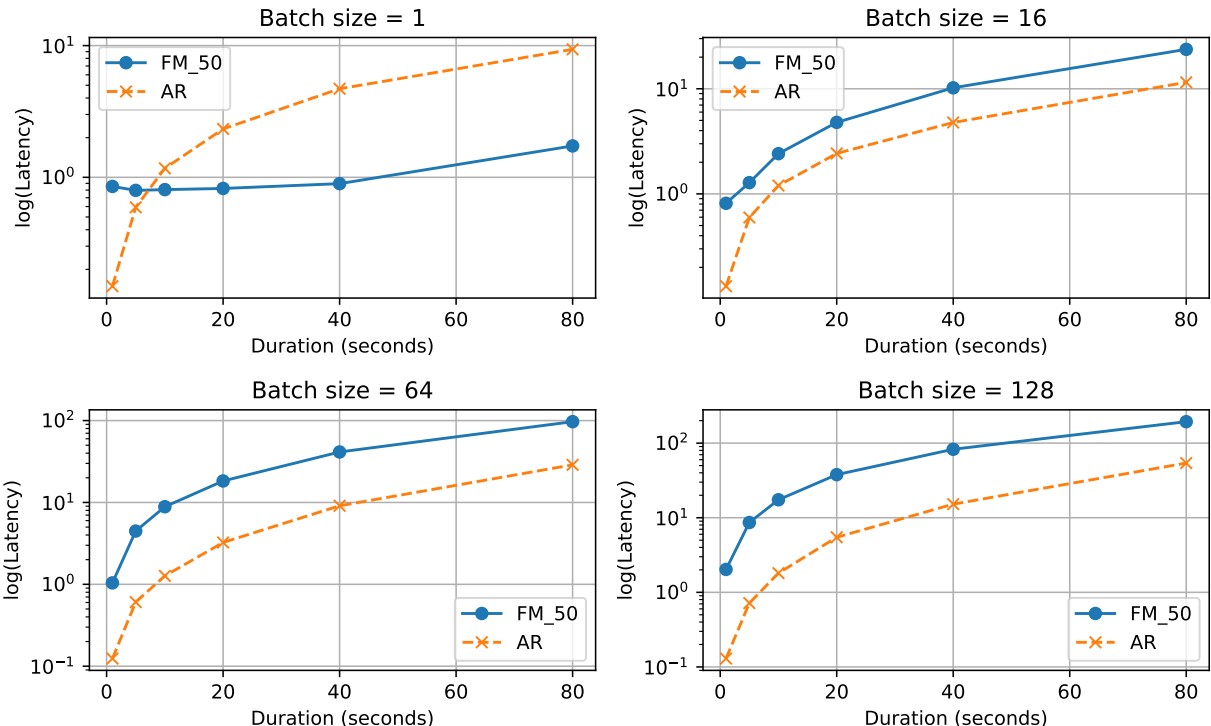

Figure 6: Measured latency for fixed batch size and extending sequence duration. Backbone contains 2 transformer layers.

evaluated with KV caching enabled. FM is evaluated with Euler's method fixed at 200, 50, 25, and 10 steps. Objective scores for these settings appear in Appendix H (Table 8). Figure 5 shows that the throughput for AR rises consistently, reaching 6.5 samples per second at batch 256. In contrast, FM plateaus: using 50 steps following Euler's solver, tops out near 3.5 samples per second. Euler's method using 10 inference steps

is faster than AR at every batch size, but the audio quality is significantly lower (FAD 4.16 versus 0.40), showing a speed-fidelity tradeoff for FM.

To better understand the behavior during inference, we start by deriving the theoretical run-time complexity expectations. Given a batch size $B$, a sequence of length $T$ and latent dimension $D$; we assume that our GPU have some fixed $C$ parallelization capability. The complexity for a full-context attention would then be $O(\frac{BDT^2}{C})$, hence for FM with $K$ sequential steps we'll get an approximated complexity of $O(K \cdot \max\{\frac{BDT^2}{C}, 1\})$; where 1 corresponds to the case in which $C$ dominates the fraction. For AR, using KV caching for self-attention would allow a $O(T)$ cost instead of $O(T^2)$, which yields a single forward pass complexity of $O(\frac{BDT}{C})$. We have to do $T$ sequential steps resulting in a sequence complexity of $O(T \cdot \max\{\frac{BDT}{C}, 1\})$. We then expect to have a $\propto K$ gap in favor of AR when $C$ does not dominate the fraction.

In practice, this is not the case in the observed setup, as AR starts to show faster inference than FM only at batch sizes $> 64$. During the derivation above, we assumed that the dominant term to dictate complexity is the self-attention cost, ignoring all other factors. It was claimed in several prior works that AR inference is often memory-bandwidth bound, not compute-bound (Jin et al., 2023; Fu et al., 2024), which we hypothesize to be the case in this setup. Figure 6 draws latency plots, fixing the batch size and extending the segment duration, considering a small transformer backbone model with 2 transformer layers. The presented latencies for this case correlate with our expectations for run-time complexity.

Figure 7 presents the same latency measurement considering 24 transformer layers for the backbone models. In this case, as we use a single A100 GPU, the larger model captures significantly more memory and the measured latencies are significantly degraded for the AR case; In correlation with Jin et al. (2023); Fu et al. (2024). These measurements suggest that in the observed setup, AR would benefit from KV-caching only for $\leq 20$ seconds segments using batch sizes $\geq 64$; FM dominating all smaller batch sizes or longer durations. To further expand this observation we we a small ablation study, incrementally increasing the number of transformer layers highlighting the accumulating latency as model grows. This experiment could be found in Appendix K.

**Take-away.** As seen in this experiment and in Appendix H, there is an apparent tradeoff between model performance and the number of inference steps for FM in the observed setup. Considering inference on a single A100 GPU, AR with KV-cache mainly benefits from scaling the batch size to $\geq 64$ for sequence durations $\leq 20$ seconds and degrades with for longer sequences due to accumulating overheads. This suggest that AR models would probably be beneficial for systems expecting large demands, e.g. integration of a generative model in social media platforms. FM demonstrated faster inference in all other cases for the observed setup.

## 5.5 Sensitivity to Training Configuration

In Section 5.1 we consider models that were trained using a one million update steps and a batch size of 256. Real-world projects often operate under lower-resource constraints. In this experiment we maintain the number of update steps fixed and vary the *tokens seen per update* by changing batch size and segment duration. The goal is to observe how each modeling paradigm's performance change as a function of batch size and segment duration. For each configuration defined by batch size $\in \{8, 16, 32, 64, 128, 256\}$ and segment duration $\in \{10, 30\}$[sec] we train two generative models using AR decoding and FM.

Figure 8 shows that FAD decreases for both paradigms as batch size increases, where the reduction is steeper for the AR model. At the largest setting (256 batch-size of 10-second clips) the AR FAD almost matches the value reached after one million updates, as seen in Section 5.1, while FM levels off earlier. When considering CLAP similarity, FM benefits steadily from larger batches, whereas the AR shows some fluctuations and relatively flattens above batch size 16. Aesthetic metrics present the opposite pattern: PQ and CE are roughly constant for FM setup, while for the AR setup they consistently improve. Notice, AR results are still below the ones obtained after one million update steps, implying there it still benefit from longer training.

**Take-away.** When the number of update steps is capped, FM reaches almost the same FAD, PQ, and CE as in the one-million-step topline using batch sizes $\geq 32$, though its CLAP score keeps improving with scale. The AR model needs a larger token budget per update step to match its topline performance, benefiting

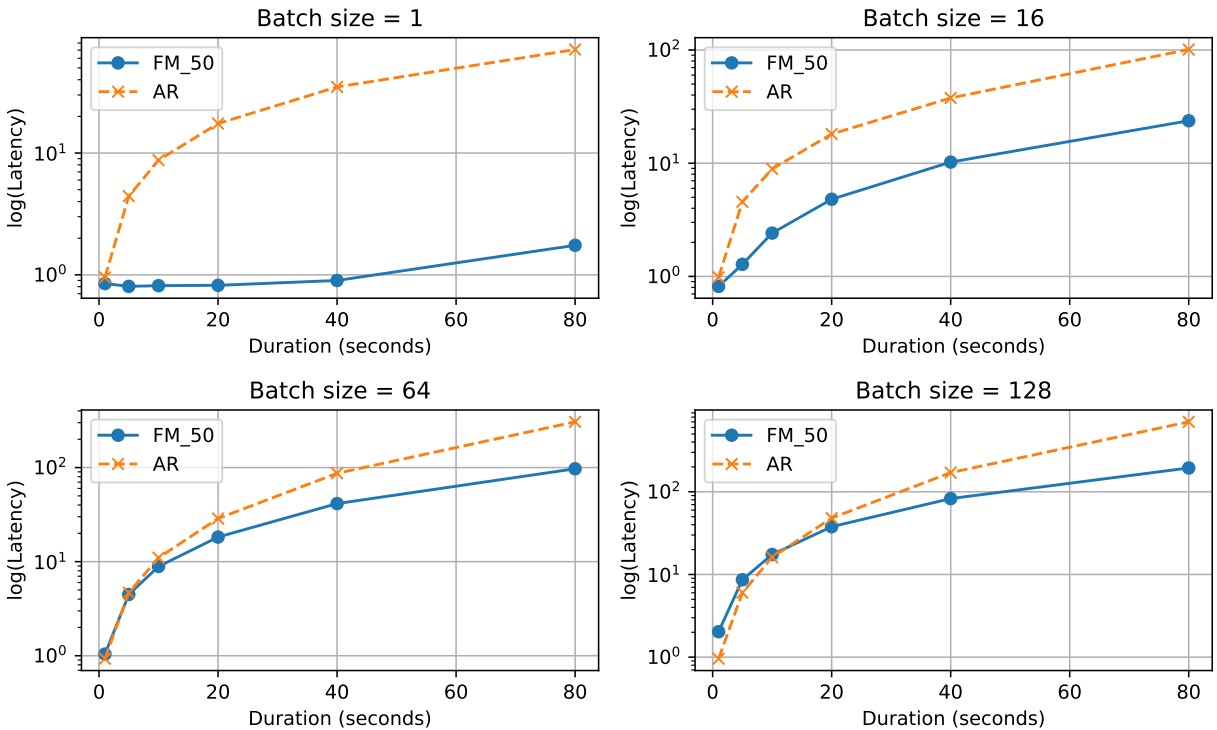

Figure 7: Measured latency for fixed batch size and extending sequence duration. Backbone contains 24 transformer layers.

more from large scale training. These observations suggest that both modeling paradigms would benefit from large-scale training, but FM could offer a more budget-friendly performance trade-off.

## 6 Conclusion

This work presents a systematic comparative study of two prominent modeling paradigms for text-to-music generation: Auto-Regressive (AR) decoding and Conditional Flow-Matching (FM). To isolate the effects of modeling choice, we fixed all other factors: training data, latent representation, backbone architecture, and evaluation protocols. We evaluated both paradigms across five axes: generation quality, control adherence, inpainting capabilities, inference efficiency, and robustness to training configurations.

**Key observations**  Our controlled experiments reveal that AR models demonstrate slightly higher perceptual quality, particularly in Fréchet Audio Distance (FAD), and better adherence to temporally aligned conditioning. FM models, on the other hand, exhibit strengths in flexibility and editing tasks, notably excelling in supervised inpainting and maintaining inference efficiency in the majority of cases. Notable trade-offs emerged across paradigms: FM requires a high number of inference steps be comparable to AR quality, and both paradigms showed a controllability–fidelity trade-off under temporally-aligned conditioning. For a summary of our conclusions please refer to Table 1,

**Limitations and future-work**  This study is centered on a single 400M-parameter transformer model and maintained a controlled experimental setup across all evaluations. While our findings reveal clear distinctions between AR and FM under these constraints, we acknowledge that alternative sampling strategies, efficient training methods, architectural innovations, and model scaling could yield different results. Future work

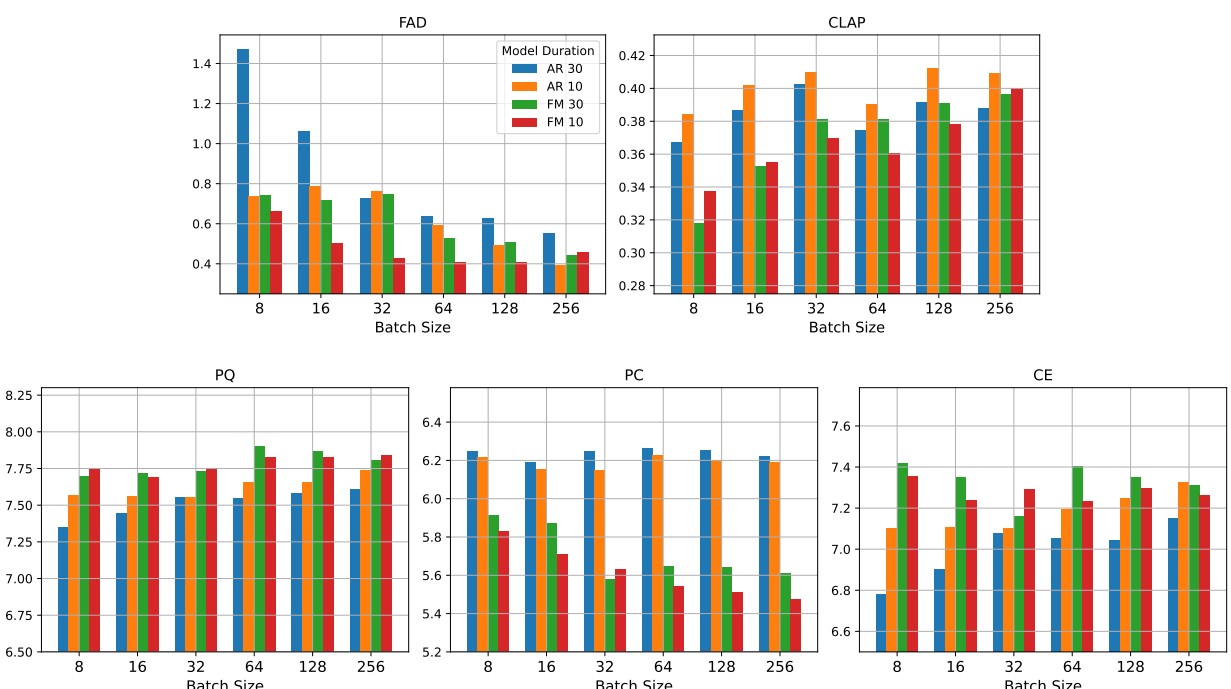

Figure 8: Objective scores after 500k updates as a function of batch size and segment duration. Both paradigms improve with more tokens per update step, but AR is more sensitive to the change.

would explore these axes to more comprehensively assess the strengths and limitations of each paradigm. We encourage the community to further investigate AR and FM models under fair, unified settings to advance our collective understanding of controllable and efficient music generation.

**Broader impact** Large scale generative models raise several ethical concerns. To address these, we ensured that all training data was obtained through legal agreements with the appropriate rights holders, primarily through a licensing agreement with the data providers. Generative models may also create challenges for artists by introducing unfair competition, which remains an open issue. We believe that open research is important for providing equal access to all participants, thus we see this research as an important milestone for sharing relevant knowledge not only for researchers in the field but also for anyone working on generative music application. By sharing insights on controlability, e.g. melody conditioning, we believe models developed considering such controls could be valuable tools for both amateur and professional musicians.

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

# A   Dataset specifications

To date, publicly available human-annotated text-to-music data is limited in terms of quantity, audio-quality, genre diversity and paired textual description quality. To the best of our knowledge - there is not a single publicly available dataset of large quantities of audio files that contains genre-diverse, high quality audio with human annotated text-descriptions. While perhaps there are existing datasets with sufficient quality and quantity (e.g. Bogdanov et al. (2019); Roy et al. (2025)) - obtaining text descriptions for the data would be done by pseudo labeling - an additional free-variable we wished to avoid in this study. To avoid that, we have sourced our data from Pond (`https://pond5.com/`) and Shutterstock (`https://shutterstock.com/music`). Using these platforms to source the data allows us control over genre diversity, having paired human-labeled descriptions and obtaining high quality audio data. As for evaluation data, the most common dataset used for evaluation is MusicCaps. This dataset's samples are fetched via download from YouTube, which may vary in encoding and sampling rates or recording conditions having a high variance and low textual-match descriptions. Measuring FAD/CLAP/Aesthetics w.r.t to this set only serves to point out large differences - where it is unclear what subtle differences stand for as this whole set is of lower quality on both audio and text match. That being said, using a high quality evaluation set serves to point out finer differences with better reliability - highlighting one method over the other. To demonstrate text-match and audio quality we employ CLAP cosine similarity and Audiobox-aesthetics metrics over the ground-truth audio and text pairs:

| Dataset | CE | PQ | CLAP |
|---|---|---|---|
| Our evaluation set | $7.23 \pm 0.57$ | $7.76 \pm 0.41$ | $0.34 \pm 0.11$ |
| pond 5 | $7.05 \pm 0.79$ | $7.79 \pm 0.48$ | $0.34 \pm 0.12$ |
| Shutterstock | $7.27 \pm 0.50$ | $7.84 \pm 0.35$ | $0.39 \pm 0.09$ |
| MusicCaps | $6.14 \pm 1.44$ | $6.91 \pm 1.17$ | $0.28 \pm 0.12$ |

From these metrics it is apparent that the audio quality suffers from a large variance, where lower PQ implies noisier recordings correlating with CE (subjective evaluation proxy). CLAP values also demonstrate a notable gap in text-audio match. These further outline the advantages of using our suggested datasets for training and evaluation as they offer higher reliability and better confidence in nuanced differences between the observed models as a result.

# B   Latent representation reconstruction comparison

We used the training recipe from the open-source implementation of StableAudio to train a VAE-gan representation using a comparable model size and the same latent frame rate, training configuration, and data in order to have an apples-to-apples comparison.

| Frame-rate | 25 | | 50 | | 100 | |
|---|---|---|---|---|---|---|
| | VAE | EnCodec | VAE | EnCodec | VAE | EnCodec |
| SISNR | 4.508 | 5.497 | 7.992 | 8.213 | 11.963 | 10.694 |
| ViSQOL | 3.859 | 3.879 | 4.062 | 4.005 | 4.191 | 4.111 |
| log-MSD | 0.414 | 0.404 | 0.339 | 0.351 | 0.279 | 0.314 |

Both models show relatively comparable reconstruction quality (SiSNR being more sensitive to phase differences); EnCodec was sampled without quantization (pre-quantizer latent was passed to the decoder)

# C   Temporal Controls Preprocessing

In this work we consider drum beat, melody, and chord progression conditioning. We perform an offline preprocessing stage for each observed latent representation frame rate, and save the preprocessed conditioning signals to memory.

**Drum Beat**   To obtain the drum beat supervision we follow  Tal et al. (2024), utilizing a pretrained EnCodec (Défossez et al., 2022) model. For each data sample, we first encode it to it's corresponding pre-quantization continuous representation using a pretrained EnCodec model that operates in the expected latent representation frame rate. We then perform temporal blurring (Tal et al., 2024), averaging every 5 sequential latent vectors and broadcast them back to their original frame rate. Finally, we pass the blurred latent vector through the first vector quantization layer of the pretrained EnCodec model, and save the resulted integer sequence to memory.

**Melody**   To obtain the melody condition we use the pretrained deep salience multi-F0 detector [10] (Bittner et al., 2017). The pretrained multi-F0 detector outputs a confidence score over a predetermined range of 53 notes (G2 to B7) spanning over $\sim$ 86Hz confidence vector sequence. Given the expected latent representation frame rate, we perform a simple linear interpolation to stretch / shrink the confidence vector stream to match it. We then pass a threshold of 0.5 confidence score, zeroing out all values below threshold. Finally, we create an integer sequence for each data sample, replacing each entry with its corresponding argmax or 54 in case the column contains only zeros.

**Chord Progression**   To obtain chord progressions, we use the Chordino [11] chord extraction model and create a (<chord label>, <switch time in sec>) pairs sequence for each data sample in our dataset. Chordino has a vocabulary size of 193 different chords, hence we create a chord to index mapping and save it to memory to be further used for tokenization. Given an expected latent frame rate, we can then convert the extracted chord sequences to integer sequences using the precomputed chord to index mapping, quantizing the <switch time in sec> timestamps to match the expected frame rate, repeating the same chord index until the next switch.

## D   Sampling Hyperparameter Search

For both modeling paradigms we experiment with classifier free guidance coefficients $\in \{1.0, 2.0, 3.0, 4.0, 5.0, 6.0, 7.0, 8.0, 9.0\}$.

For FM we do inference with Dopri (dynammic number of steps) or Euler (considering number of steps: $\in \{10, 25, 50, 150.250\}$).

For AR we explore temperature $\in \{1.2, 1.4, 1.6, 2.0, 2.4, 2.8\}$, top p $\in \{0.6, 0.8\}$ or top k $\in \{250, 500\}$.

## E   Latent representation and transformer model specifications.

### E.1   Latent Representation

We follow the approach taken in Copet et al. (2023); Vyas et al. (2023); Tal et al. (2024) and use En-Codec's (Défossez et al., 2022) quantized discrete representation for AR modeling and its continuous, pre-quantizer, latent for FM modeling. Using un-normalized representations shows a notable performance gap. In addition, we train a VAE-GAN autoencoder following StableAudio's (Evans et al., 2025) open source recipe[12], training with a latent KL divergence constraint w.r.t $\mathcal{N}(0, I)$, without any quantization performed during training. We train $\{25, 50, 100\}$[Hz] latent frequency variants for each model, training for $400k$ steps using AdamW optimizer with a learning rate of $3 \cdot 10^{-4}$ considering $\sim$ 1[Sec] segments and a batch size of 64 samples. The specific configurations used for each model variant can be found in subsection E.3.

### E.2   Normalizing EnCodec Latent Representation For FM modeling.

We sample $N = 2048$ 10 second random segments from our train set and encode them to a latent representation matrix $M$ of shape $[N, T, D]$ where $D$ is the latent dimension and $T$ is the corresponding temporal dimension $T = 10 \cdot f_r$. We compute a single scalar for the empirical mean and empirical mean std as follows:

---

[10]`https://github.com/rabitt/ismir2017-deepsalience`
[11]`https://github.com/ohollo/chord-extractor`
[12]`https://github.com/Stability-AI/stable-audio-tools/tree/main`

```
mean = M.mean()
mean_std = M.std(dim=1).mean()
---
def normalize(z: Tensor):
    return (z - mean) / mean_std

def unnormalize(z: Tensor):
    return z * mean_std + mean
```

### E.3 Latent Representation Models Configurations

Both of our observed latent representation models are symmetric auto-encoder models. We use the open-source implementation in Audiocraft to train EnCodec, and the implementation in Stable-Audio-Tools to train the VAE. The table below specifies the critical hyper-parameters required to train each of the representation model configurations. In the Discrete EnCodec case we use 4 residual codebooks, each containing 2048 bins.

| Model | Frame Rate | Strides | Channels | Activation | Latent Dimension |
|---|---|---|---|---|---|
| EnCodec | 25
50
100 | [8, 8, 5, 4]
[8, 5, 4, 4]
[8, 5, 4, 2] | [64, 128, 256, 512] | GELU | 128 |
| VAE
$\alpha_{\mathrm{KL}} = 10^{-3}$ | 25
50
100 | [2, 4, 4, 5, 8]
[2, 4, 4, 4, 5]
[2, 2, 4, 4, 5] | [128, 256, 512, 1024, 2048] | Snake | 64 |

#### E.3.1 Backbone Transformer

For the backbone transformer decoder model we follow the implementation of Copet et al. (2023) using a 400M parameters configuration containing 24 transformer decoder blocks with a hidden dimension of 1024, 16 multi-head attention layers and a feed-forward dimension of 4096. We use T5 (Raffel et al., 2020) to obtain text embeddings and pass them via cross-attention layers as text conditions. For the FM case we perform slight modifications similarly to Tal et al. (2024) and include U-Net-like skip connections. With $2N$ being the number of transformer decoder blocks, we add skip-connections, connecting the input of the $i$'th block with the $2N - i$'th block output, for $i \geq N + 1$. Each skip connection follows a simple concatenation and linear projection: Linear(Concat($x$, skip)). The skip connections add $\sim 7$M parameters to the network and the input projection for the temporal conditioning experiments add $\sim 1$M parameters. The FM transformer model doesn't require the input embedding tables, reducing $\sim 8$M parameters.

## F Fixed training setup evaluation over MusicCaps

Re-evaluating the experiment presented at Table 2 on MusicCaps (Agostinelli et al., 2023) dataset yields the results depicted in Table 6. The results show a similar trend to Table 2 where AR performs better than FM in most cases, where there is less consistency regarding the representation used for FM or the degradation of performance as frame rate increases. As shown in Appendix B, MusicCaps suffers from high variance in quality and text-audio match in comparison to our chosen evaluation set - hindering the reliability of the observed trends in this experiment. Therefore, one could view this experimentation as a relative reference point to performance demonstrated by other music-generation works. For example, MusicGen's small configuration had reported to have 3.1 FAD score using a similar evaluation setup.

Table 6: Fixed training configuration evaluation over MusicCaps

| FR | Modeling | FAD↓ | CLAP↑ | PQ↑ | PC↑ | CE↑ |
|---|---|---|---|---|---|---|
| | AR | 4.10 | 0.33 | 7.17 | 5.13 | 6.59 |
| 25 | FM | 5.53 | 0.30 | 7.42 | 4.17 | 6.42 |
| | FM (VAE) | 4.58 | 0.30 | 6.99 | 4.70 | 6.24 |
| | AR | 3.55 | 0.33 | 7.10 | 4.93 | 6.36 |
| 50 | FM | 5.65 | 0.29 | 7.39 | 4.26 | 6.28 |
| | FM (VAE) | 5.42 | 0.29 | 7.10 | 4.60 | 6.10 |
| | AR | 3.52 | 0.33 | 6.63 | 5.27 | 6.39 |
| 100 | FM | 6.02 | 0.29 | 7.47 | 4.53 | 6.45 |
| | FM (VAE) | 4.62 | 0.29 | 7.00 | 5.54 | 6.65 |

## G   Exploring sensitivity to latent frame rate

The full evaluation scores for the latent frame rate ablation in Section 5.1. Observations shows that both sequence length and the latent representation itself affect performance, with the latter being more crucial.

Table 7: Exploring sensitivity to latent frame rate

| FR | Dur (s) | FAD↓ | CLAP↑ | PQ↑ | CE↑ |
|---|---|---|---|---|---|
| | 10 | 0.42 | 0.39 | 7.78 | 7.13 |
| 25 | 20 | 0.45 | 0.39 | 7.76 | 7.16 |
| | 40 | 0.51 | 0.38 | 7.74 | 7.11 |
| | 10 | 0.48 | 0.40 | 7.73 | 7.20 |
| 50 | 20 | 0.62 | 0.40 | 7.65 | 7.12 |
| | 40 | 0.65 | 0.39 | 7.69 | 7.23 |
| 100 | 10 | 0.68 | 0.38 | 7.37 | 7.10 |

## H   Constricting Sampling Steps

Table 8 depicts the full tradeoff w.r.t objective quality metrics when fixing the number of sampling steps made with Euler ODE solver during FM inference.

## I   Entropy Analysis Under Strict Temporal Conditioning

To shed light on the quality drop observed in Table 3, we measure how strict conditioning affects the uncertainty of the AR decoder. For each test prompt we record the token probability over the first EnCodec codebook stream at every sampling step and compute its entropy. The curve is averaged over 100 random prompts; the text-to-music baseline is obtained by using the models trained in Section 5.1.

Figure 9 shows that conditioning sharpens the distribution early in the sequence, where the provided control tokens directly determine the next few audio tokens, but then the entropy plateaus around timestep 50. We hypothesize that the conditioning imposes a strong bias and therefore leads to a higher chance of sampling out-of-distribution token sequences (some steps doesn't match the conditions), and causes a higher entropy

Table 8: FM (EnCodec) sampling using fixed number of steps with Euler ODE solver.

| Hz | Steps | FAD↓ | Clap↑ | PQ↑ | PC↑ | CE↑ |
|---|---|---|---|---|---|---|
| | 200 | 0.45 | 0.39 | 7.73 | 5.50 | 7.16 |
| 25 | 50 | 0.74 | 0.41 | 7.76 | 5.63 | 7.24 |
| | 25 | 0.99 | 0.39 | 7.54 | 5.48 | 7.01 |
| | 10 | 4.16 | 0.27 | 6.59 | 5.07 | 5.94 |
| | 200 | 0.90 | 0.39 | 7.49 | 5.70 | 6.97 |
| 50 | 50 | 1.33 | 0.40 | 7.45 | 5.68 | 6.95 |
| | 25 | 1.87 | 0.38 | 7.29 | 5.64 | 6.78 |
| | 10 | 4.92 | 0.30 | 6.70 | 5.46 | 6.10 |
| | 200 | 0.81 | 0.39 | 7.40 | 5.64 | 6.86 |
| 100 | 50 | 1.18 | 0.39 | 7.17 | 5.74 | 6.72 |
| | 25 | 2.91 | 0.35 | 6.73 | 5.7 | 6.31 |
| | 10 | 12.3 | 0.20 | 5.60 | 5.02 | 4.67 |

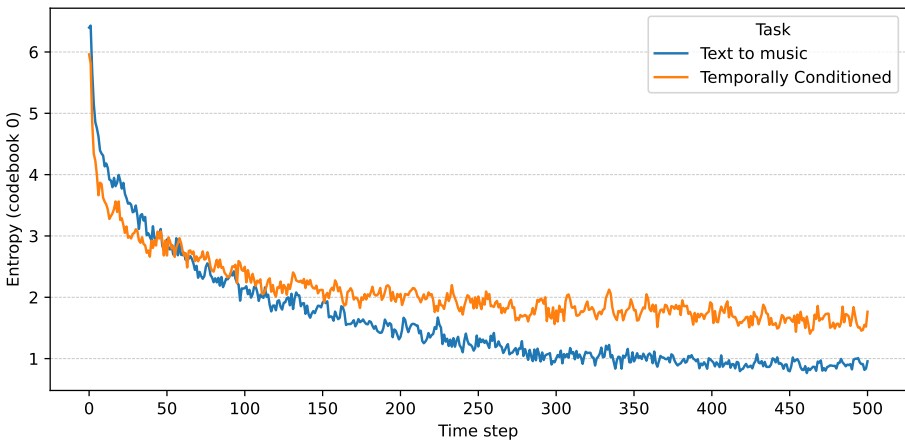

Figure 9: Average entropy of the AR decoder as a function of sampling step. Conditioning lowers entropy in the first ∼50 steps, indicating confident early predictions, but raises it later, suggesting increased sensitivity to low-probability paths.

as the sequence becomes longer due to that. This manifests as the FAD and CLAP degradation reported in Section-5.2. This also implies that using multi-source classifier-free guidance (Tal et al., 2024) may mitigate this effect, and we leave that for future work.

## J   Inpainting algorithms

In the fine-tuning case of AR decoding, we need to introduce 3 new tokens (<a>, , <c>) in order to partition the source latent representation to 3 segments: A,B,C. B is the segment to be inpainted. We place the special token prior to each segment, resulting in a $T + 3$ length segment, and reorganize the sequence as [<a>, A, <c>, C, , B]. During inference we give [<a>, A, <c>, C, ] as a prompt to the model, which continues to generate until <eos> or max generation length is met. We then re-organize the segments to A, B, C and reconstruct the waveform.

---

**Algorithm 1** Zero-Shot Inpainting Evaluation via Flow Matching Inversion

---

**Require:** Source latent $\mathbf{z}_0$ which is a sequence of $T$ latent vectors, condition tensors $\mathcal{C}$, guidance terms $\mathcal{G}$, number of Euler steps $N$ and a fixed mask size $M = T/2$
**Ensure:** Inpainted latent $\mathbf{z}$

1: **Initialize:** $\Delta t \leftarrow 1/N$, $\mathbf{z} \leftarrow \mathbf{z}_0$, $t \leftarrow 1$
2: Create empty list `noises` $\leftarrow [\,]$                 $\triangleright$ Perform inversion
3: **for** $i = 1$ to $N$ **do**
4:      Compute $\mathbf{v}_\theta \leftarrow \text{model}(\mathbf{z}, t, \mathcal{C}, \mathcal{G})$
5:      Update $\mathbf{z} \leftarrow \mathbf{z} - \Delta t \cdot \mathbf{v}_\theta$
6:      Append current $\mathbf{z}$ to `noises`
7:      Update $t \leftarrow t - \Delta t$
8: **end for**
9: Reverse the list `noises`
10: **Sample** random start index $s \sim \text{Uniform}(0.1 \cdot T, 0.9 \cdot T)$
11: Set $e \leftarrow s + M$
12: Initialize $\mathbf{z} \sim \mathcal{N}(0, I)$, $t \leftarrow 0$             $\triangleright$ Perform inpainting
13: **for** each noise $\mathbf{n}$ in `noises` **do**
14:      Replace $\mathbf{z}[:, : s] \leftarrow \mathbf{n}[:, : s]$, $\mathbf{z}[:, e :] \leftarrow \mathbf{n}[:, e :]$
15:      Compute $\mathbf{v}_\theta \leftarrow \text{model}(\mathbf{z}, t, \mathcal{C}, \mathcal{G})$
16:      Update $\mathbf{z} \leftarrow \mathbf{z} + \Delta t \cdot \mathbf{v}_\theta$
17:      Update $t \leftarrow t + \Delta t$
18: **end for**
19: Replace $\mathbf{z}[:, : s] \leftarrow \mathbf{z}_0[:, : s]$, $\mathbf{z}[:, e :] \leftarrow \mathbf{z}_0[:, e :]$
20: **return z**

---

**Algorithm 2** Supervised Inpainting Flow Matching

---

**Require:** Source latent $\mathbf{z}_0$, condition tensors $\mathcal{C}$, guidance terms $\mathcal{G}$, number of Euler steps $N$ and a fixed mask size $M = T/2$
**Ensure:** Inpainted latent $\mathbf{z}$

1: **Sample** random start index $s \sim \text{Uniform}(0.1 \cdot T, 0.9 \cdot T)$
2: Set $e \leftarrow s + M$
3: **for** step in $\{1, ..., N\}$ **do**
4:      Replace $\mathbf{z}[:, : s] \leftarrow \mathbf{z_0}[:, : s]$, $\mathbf{z}[:, e :] \leftarrow \mathbf{z_0}[:, e :]$
5:      Compute $\mathbf{v}_\theta \leftarrow \text{model}(\mathbf{z}, t, \mathcal{C}, \mathcal{G})$
6:      Update $\mathbf{z} \leftarrow \mathbf{z} + \Delta t \cdot \mathbf{v}_\theta$
7:      Update $t \leftarrow t + \Delta t$
8: **end for**
9: Replace $\mathbf{z}[:, : s] \leftarrow \mathbf{z}_0[:, : s]$, $\mathbf{z}[:, e :] \leftarrow \mathbf{z}_0[:, e :]$
10: **return z**

---

For FM, we could perform inpainting either by using a pretrained model and perform zero-shot inpainting, or by training a model specifically for the task. Algorithm 1 describes the naive logic implemented to allow for zero-shot inpainting using a FM model and a fixed sampling schedule. In the supervised case of FM, we do not perform inversion, and simply plug in the latent representation itself in the unmasked segments as depicted in algorithm 2.

## K   Empirical observation over GPU-Memory utilization

To better capture the the model size - AR inference speed tradeoff presented in Section 5.4, we perform a controlled ablation study, considering $[2, 4, 8, 16, 24]$ transformer layers in the backbone models in order to observe the tradoff transitioning trends. To better isolate the runtime overheads, we discard cross attention layers (that can't use KV-cache) and replace the AR softmax + top-p sampling with argmax sampling.

Figure 10 clearly demonstrate the shift in AR latency, from dominating FM considering 2 layers to exponentially exceeding FM latency. As expected, longer sequences seem to be increasing in larger latency strides as the performance gap accumulates with $T \gg K$. Interestingly, despite suffering from large degradations as model size increase - increasing the batch size keeps improving throughput ($\frac{\text{batch size}}{\text{latency}}$) yet with apparent diminishing returns as model size increases, further highlighting the implications of less available memory.

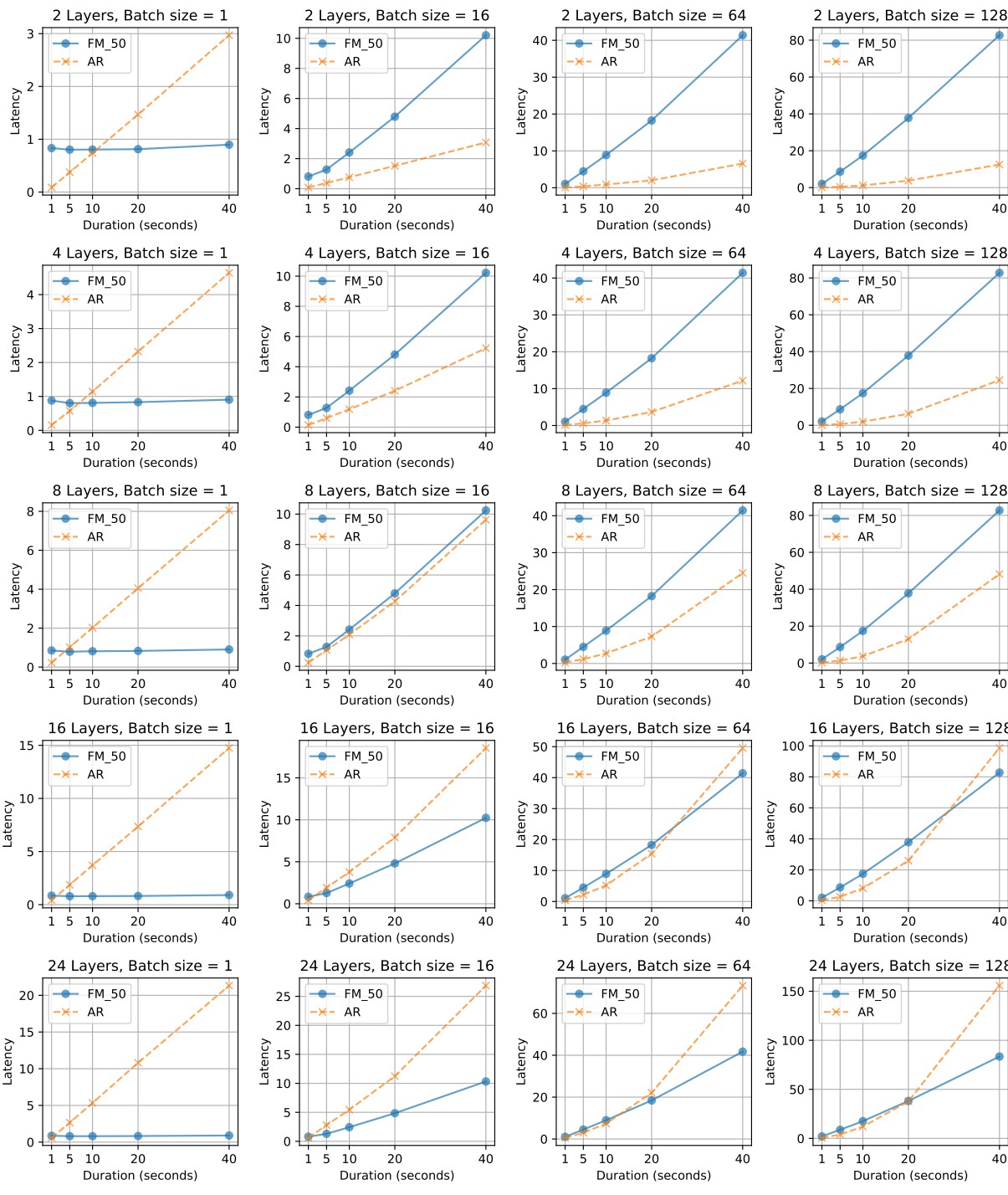

Figure 10: Observing the impact of incrementing the number of transformer layers on latency. Column shift from left to right capture batch size increase 2 → 128, and row shift from top to bottom represent increase in the number of layers.

