# OpenReview forum: "Auto-Regressive vs Flow-Matching: a Comparative Study of Modeling Paradigms for Text-to-Music Generation"
_TMLR — Accepted by TMLR_

### Review · Reviewer_LfSu · 2025-06-25

**Summary Of Contributions:**

This paper presents a comparative study of autoregressive (AR) models and conditional flow matching (CFM) models for the task of text-to-music generation. In a controlled experiment setting, the authors compared these two approaches in terms of temporal alignment, objective quality, temporal control adherence, inpainting quality, runtime, model scaling, and sensitivity to batch sizes.

**Audience:**

Yes

**Claims And Evidence:**

Yes

**Requested Changes:**

Can the authors include some more discussions on the choice of the tokenizers?

**Strengths And Weaknesses:**

Strengths:

- The paper is neatly written with a clear motivation and objective to compare AR and flow-matching models in the context of text-to-music generation. This paper would serve as a useful reference for researchers in this area.
- The experiments are carefully constructed and controlled, providing clean signals and takeaways.

Weaknesses:

The tokenizers play an important role in generation, but their effects are not carefully studied. A discrete tokenizer (EnCodec) is used for the AR model. The pre-quantized version of EnCodec is for the flow-matching model, in addition to a variant trained with continuous latents of a VAE. The VAE model seems to be clearly worse (perhaps there could be better continuous latents? e.g., scaling the VAE model size or latent dimensions).

As for using EnCodec, it is unclear whether the pre-quantized features will act as good continuous features for generative modeling. For example, the features might be less smooth compared to the features generated by a VAE due to the discrete training procedure. In other words, the performance differences shown in the experiments might be due to the choice of the tokenizer.

Minor comment: In the first paragraph, the authors mentioned that the generation methods in text/images have converged. I don’t agree with this because there are continuous and recent developments in discrete flow matching text generation and autoregressive image generation.

---

> ### Author Response · Authors · 2025-08-06
>
> Dear reviewer,
> We thank you for taking the time to review our work and for the thoughtful feedback.
> Please see our comments below.
>
> **Convergence wording.** We agree that “convergence” overstates the current landscape. We clarified that in the first paragraph of the introduction in the revised paper.
>
> **Latent representation.** We agree that tokenization can affect downstream performance.
> In this study we aim to isolate the modeling paradigm as much as possible. To do so we use a single representation for both AR and FM. We chose EnCodec as it is widely used by several works for both AR and FM e.g. [MusiConGen](https://arxiv.org/pdf/2407.15060v1), [CoCo-MuLLa](https://arxiv.org/abs/2310.17162), [AIRGen](https://arxiv.org/pdf/2402.09508), [MusicFlow](https://arxiv.org/pdf/2410.20478).
> For completeness, we additionally compare EnCoded to a VAE-based representation. We use the training code from the open-source implementation of [StableAudioOpen](https://arxiv.org/pdf/2407.14358) to train a VAE-gan representation using a comparable model size and the same latent frame rate, training configuration and data to have an apples-to-apples comparison.
> The VAE-gan has comparable reconstruction quality as EnCodec (w.o quantization) but it underperformed when used as a representation space for generative tasks, as seen in the results of section 5.1.
> We included clarifications and an expanded discussion of these limitations in the revision.
>
> |||25 Hz||50 Hz||100 Hz||
> |-|-|-|-|-|-|-|-|
> |||VAE|EnCodec|VAE|EnCodec|VAE   |EnCodec|
> |SISNR||4.508|5.497|7.992|8.213|11.963|10.694|
> |ViSQOL||3.859|3.879|4.062|4.005|4.191|4.111|
> |log-MSD||0.414|0.404|0.339|0.351|0.279|0.314|
>
> **Manuscript updates.** We:
> 1. Revised the first paragraph of the introduction as above;
> 2. Edited section 4 (Input Representation)
> 3. Expanded the conclusions to acknowledge that alternate latents, sampling schemes, architectures, and scaling may yield different trade-offs and merit future investigation.
>
> We appreciate the constructive critique and we hope these clarifications address the reviewer’s concerns

---

### Review · Reviewer_24at · 2025-07-01

**Summary Of Contributions:**

This paper presents a comparative study of recent text-to-music generation methods motivated by the growing interest in this field and the increasing diversity of approaches which currently lack a unified framework for fair evaluation. While factors such as datasets or architecture design choices may significantly influence the performance of these models, this study only focuses on the modeling paradigm and compares Auto-Regressive (AR) vs Flow-Matching (FM) methods, excluding recent hybrid models.

The main contribution lies in the empirical comparison of both AR and FM methods relying on existing works, on 5 aspects: generation fidelity, adherence to temporal controls, inpainting ability, inference and training configurations (including inference speed, batch size and segment duration). Both quantitative and qualitative analysis are conducted. They highlight the trade-off between controllability and generation quality. The AR method appears to yield slightly better results except for the inpaintaing task.

**Audience:**

Yes

**Broader Impact Concerns:**

The paper presents practical insights of state-of-the-art text-to-music generation methods to guide future research and technical improvements. It does not address the broader societal or ethical implications of such technologies but as text-to-music systems improve in quality and accessibility, they raise several concerns that may be mentioned.

**Claims And Evidence:**

Yes

**Requested Changes:**

Some key points to consider:

- Sec. 2. in related work, mention prior studies and clearly state your contributions
- Sec. 4.1 better justify the motivation for using a private proprietary dataset
- Sec. 5.1 Is there a reason why the FM approach is very sensitive to the latent frame rate?
- Sec. 5.1 The paper states that “The best performing FM configurations follows the Dopri5…”  but the results for this are not provided to compare with Euler’s method
- Sec. 5.2 Why not also comparing with the FM VAE-based approach?
- Sec. 5.5 Why not training for 1M steps as in Sec.5.1 and give the results at 500k steps if the model is not completely trained yet? (also FAD for AR is higher than for FM which may be a bit confusing with the results from Sec. 5.1)

List of minor adjustments:

- Sec. 1. Introduction may also mentioned speech related works (only citing NLP —> AR, Computer Vision Non-AR)
- Sec. 1. Introduction: “We evaluate each modeling paradigm across multiple
axes including perceptual quality, inference efficiency, robustness to training configuration, adherence to temporal control, and editing capabilities in the form of audio inpainting.” —>  can indicate each related sections and refer to Table 1 here while keeping the same order to improve readability
- some references need to be fixed: (Evans et al. 2024) —> ICASSP 2024, (Lipman et al. 2022) —> ICLR 2023
- Sec. 2. Related work: “AudioLM … AR Transformer to predict them. This enable …” —> enables
- Sec. 3.3 Please write equation (3) more rigorously (E_{t \sim [0, 1] …}) ; remove “:” before equation (5)
- Table 2 some results could be highlighted to improve readability
- Sec. 5.4 “we show that FM can approach AR quality … ” —> can “closely match” or “achieve”
- Sec. 6. Conclusion “On the compute side AR …” —> “On the computational side …”

**Strengths And Weaknesses:**

The paper addresses an important topic as text-to-music generation is advancing rapidly with a growing variety of approaches, and a unified evaluation framework is essential for guiding future research and supporting meaningful progress and fair comparison across models. The proposed evaluation method is well structured and covers multiple practical aspects. The paper is clearly written and well organized. The experimental setup is clear and the inclusion of audio examples on the supporting webpage is highly appreciated and adds valuable insight.

However, while the paper is generally well-written and provides a comprehensive description of the methods, the results section could benefit from deeper analysis and interpretation. A more thorough discussion of the findings (beyond reporting metrics) would help clarify the implications of the comparisons and strengthen the overall contribution.

Additionally, the related work section does not mention any prior studies or comparative analysis of these approaches which makes it difficult to clearly identify the novelty and key contributions of this study compared to prior works.

Finally, although datasets are not the focus of this study, the use of a private proprietary dataset may hinder the reproducibility of the results and the motivation for choosing this dataset is not clearly explained (Sec. 4.1 ”we believe…”).

---

> ### Author Response · Authors · 2025-08-06
>
> Dear reviewer,
> We thank you for taking the time to review our work and for the thoughtful feedback.
> We will revise the mentioned minor issues, and include a broader impact paragraph in section 6.
> Please see our comment below.
>
> **Prior work & our contributions.** With the field evolving rapidly, most prior comparative studies are surveys highlighting progress made via both auto-regressive and non-auto-regressive models. We cite several such works early in the introduction (e.g. https://arxiv.org/pdf/2308.12982).
> Many individual generative works adopt one modeling paradigm and compare to others, but typically vary in architecture, data, or other confounding factors. To our knowledge, this is the first direct comparison of modeling paradigms in music generation under controlled conditions. In our Related Work section, we focus on how AR and non-AR approaches have each progressed, motivating our study. Since our work is comparative, the main contributions are the derived conclusions. We summarize them clearly at the end of the introduction (Table 1). We would be happy to include any missing references upon the reviewer’s request.
>
> **Using a proprietary dataset.** Public text-to-music datasets remain limited in size, diversity, and annotation quality. To our knowledge, no public dataset offers both large-scale, genre-diverse, high-quality audio and human-written captions. Hence, we use licensed datasets from [Pond5](https://pond5.com/) and [Shutterstock](https://shutterstock.com/music); these sources of data were mentioned by several prior works. Using these platforms to source the data allows us control over genre diversity, having paired human-labeled descriptions and obtaining high quality audio data.
> As for evaluation, MusicCaps is commonly used, but suffers from inconsistent quality due to being sourced from YouTube. Its variability in encoding and loose text alignment makes it unreliable for subtle comparisons. Thus, we chose a high-quality proprietary test set to ensure more meaningful differentiation between models.
> To support the above claims, we report CLAP similarity and Audiobox-Aesthetics on ground-truth audio-text pairs:
> |Dataset|CE|PQ|clap|
> |-|-|-|-|
> |our eval|7.23±0.57|7.76±0.41|0.34±0.11|
> |p5|7.05±0.79|7.79±0.48|0.34±0.12|
> |sttrstk|7.27±0.50|7.84±0.35|0.39±0.09|
> |musiccaps|6.14±1.44|6.91±1.17|0.28±0.12|
>
> These show that MusicCaps has higher variance and lower fidelity. PQ correlates with CE (subjective perception), while CLAP highlights audio-text mismatch. Overall, our dataset choices allow for more reliable, fine-grained evaluation of modeling approaches.
>
> **Reason for FM being more sensitive to frame rate.** This is a great question to be explored in depth. In our experimentation we empirically see a degradation in performance across all metrics, especially notable when using VAE. We hypothesise that the reason for that observation could stem from one of the following: (i) Each frame-rate has a different representation model - one latent representation could be more convenient to model than the other; and (ii) different lengths - modeling longer dependencies could be harder. To better understand the nature of this observation we repeat the experiment for {25, 50}Hz FM (EnCodec) models considering {20, 40}sec durations. Under this setup we could both isolate the impact of sequence length or the impact of latent frame rate.
> |FR|Dur(s)|FAD↓|PQ↑|CE↑|
> |-|-|-|-|-|
> |25|10|0.42|7.78|7.13|
> ||20|0.45|7.76|7.16|
> ||40|0.51|7.74|7.11|
> |50|10|0.48|7.73|7.20|
> ||20|0.62|7.65|7.12|
> ||40|0.65|7.69|7.23|
> |100|10|0.68|7.37|7.10|
>
> When fixing the latent frame rate, extending the sequence length has a consistent impact on performance, although the impact is more apparent at 50Hz than at 25Hz. On the other hand, considering the following pairings: {(25Hz, 20s), (50Hz, 10s)}, {(25Hz, 40s), (50Hz, 20s), (100Hz, 10s)} - we fix the length of the input sequence; showing a notable gap w.r.t change in representation frame rates. Both of these observations imply that longer sequences could have an impact on performance, results suggest that the representation itself serves as the more crucial factor of the two. We hypothesize that such implications stem from the nature of the more compressed representation, where less vectors correspond to the same temporal span - resulting in fewer local dependencies. We will include this experimentation as a part of the analysis of section 5.1.
>
> We appreciate the constructive critique and we hope these clarifications address the reviewer’s concerns.

---

### Review · Reviewer_eTk3 · 2025-07-23

**Summary Of Contributions:**

The contributions of this work is an empirical study on text-to-music generation methods is presented comparing auto-regressive and flow-matching modeling paradigms. The work uses the same training data/data pipeline and similar (but not the same) model architectures together with a short suite of tasks (text-condition generation, inpainting, time-conditioned controllable generation, inference-speed, and sensitive to training configuration) to benchmark.  Results suggest AR models are slightly preferred for text-conditioned generation, time-conditioned generation and sensitivity to training configuration and FM is preferred for inpainting and speed.

**Audience:**

Yes

**Broader Impact Concerns:**

For broader impact concerns, it's unclear to me of these conclusions will change if the model size is scaled up, if the length of generation is increased beyond 10 seconds, if the training batch size is higher (very common for diffusion/flow-matching models). This said, however, I do not doubt that the conclusions for the models detailed in the paper are accurate.

**Claims And Evidence:**

Yes

**Requested Changes:**

Overall, if the narrative and takeaways can be updated to focus more on the “actionable insights” and reproducible scientific insight and less on trying to determine the ultimate winner, I’m supportive of accept! The goal here is to not require an extreme number of additional experiments, but to help frame the narrative to make it more valuable long-term.

Ideally, the authors can also help readers have some understanding of how the models compare by comparing with pre-trained open-source models and evaluating on open-source datasets.

In the extreme case, it would also be ideal to make the model architectures between the two models the same (they are unnecessarily different now and it’s unclear why).

**Strengths And Weaknesses:**

Strengths

It’s very nice to have a deeper empirical study to compare AR and FM models on several tasks and the same dataset. The writing is very clear as well as the overall organization/structure of the work. The proposed tasks and experimental design tasks and metrics are suitable.

Areas for improvement

•	Focus on actionable insights over what model “wins” -- Focus on “actionable insights that can inform future architectural and training decisions” and not trying to decide if AR or FM wins. In the abstract, you focus on the former, but in the introduction and most of the remaining paper commentary feels more focused on trying to decide if AR or FM wins.

It will be near impossible to try to satisfy all reviewers/readers on how to make a fair comparison between the two approaches as they are fundamentally different on many axes. Furthermore, different methods might require different training properties that were normalized here (e.g. batch size), the scaling is relatively small compared to recent models (will the conclusions change with scale?), and even the basic model architecture between the two approaches was different. In contrast, however, if you focus on deeper insights and try to understand and why different outcomes were found, this will likely be much more helpful for future readers and will last the test of time longer.

Thus, for each “take-away” paragraph, try to focus not on the final winner, but on the reasons why and hypothesize how to address differences.

•	Comparison to open-source models –It would be ideal run eval using pre-trained open-source models and such as Stable Audio Open and Music Gen using your eval set and also run your models on open-source data eval sets (e.g. Song Describer). I’m sure the data you use is better than this, but the point is for a common benchmark. If the data is not good, run it, show it, and explain why, etc. While I realize this can be a complicated issues w.r.t., data licensing and that could be the limiting factor. So, if that is the case, please note it, etc.

•	Model architecture differences. I’m trying to understand why you did not use the same model architecture for your AR and FM models. Why was the AR model arch modified from the MusicGen arch to be more different than the FM model arch baseline? I think this is one of the more unclear issues for the comparison. Please add a reasonable explanation on why this is the way it is or potentially update the experiments to address (ideally this is not necessary, or could be shown in a small, separate experiment).
•	Other Minor Issues. Please see below for more minor, specific comments.

Overall, if the narrative and takeways can be updated to focus more on the “actionable insights” and reproducible scientific insight and less on trying to determine the ultimate winner, I’m supportive of accept! The goal here is to not require an extreme number of additional experiments, but to help frame the narrative to make it more valuable long-term.
Low-level Comments:
•	The abstract is clearly written and with clear purpose, ending with “providing actionable insights that can inform future architectural and training decisions in the evolving landscape of text-to-music generation.”
•	The last paragraph of the intro gives the impression of changing the clear purpose of the abstract to focus solely on the results with “Our results highlight consistent differences between the two paradigms.” There is note of “practical guidance for selecting modeling paradigms in future” but it is unclear if the guidance is simply going to be AR or LM are best or something deeper.
•	In Section 2, it would be useful to make clear which methods can natively handle stereo generation such as Stable Audio.
•	In Section 2, when introducing flow-based models, it would be beneficial to the reader to more clearly connect the similarities between diffusion and flow-matching. Right now, the difference between diffusion and flow models is presented to be similar to the difference between AR and diffusion models, but there is a much more intimate connection between diffusion and flow-models. See for example https://arxiv.org/abs/2301.12003 and its relation to EDM https://arxiv.org/abs/2206.00364 For continuous-time diffusion models like EDM, the practical implementation of flow-models becomes strikingly similar to diffusion models and this closeness would be useful to clarify.
•	In Section 2, the comment “AR models continue to set strong baselines for musical structure and coherence” please give a reference. Past literature such as noted in  Stable Audio found that diffusion (highly related to FM) gave improved structure via qualitative analysis.
•	For Section 3.1, please explain why you are focusing on generating 32kHz mono audio instead of 44.1kHz or higher stereo audio given that past methods already generate higher fidelity content. If it’s a result of licensed data access, that is totally fine, but it should just be clear to the reader since you are trying to should what method is better and immediately reducing the quality compared to previously published results (e.g. Stable Audio).
•	In Section 3.2, it would be useful to try to note some of the design decisions of AR generation models such as mentioning the complication of using a delay pattern for prediction.
•	In Section 3.3, please edit the comment “Unlike diffusion models, which rely on stochastic noise schedules and iterative denoising” Diffusion models can have deterministic samplers as well (e.g. https://arxiv.org/abs/2206.00364).
•	In Section 3.3, it would be useful to comment on the similarities with diffusion models. While the theoretical justification was developed differently, the practical implementations of training and inference are extremely similar. This will help your goal of “providing actionable insights” rather than a benchmark.
•	In Section 4.1, similar to before, please explain why you only use mono 32kHz data.
•	In Section 4.1, instead of explicitly not evaluating on MusicCaps, it would likely be more beneficial to evaluate on MusicCaps or Song Describer, show the results, and then explain why there are problems (I believe there are problems). Right now, however, it is more-or-less written as “just trust us”. To provide actionable insights, showing the performance on a standard eval benchmark is still useful.
•	In Section 4.2, please explain why you didn’t compare using the pre-trained model of . The open-source version of this implementation has a latent sampling rate of approx. 22Hz for stereo 44.1kHz audio, but you don’t use this rate, etc.
•	In Section 4.2, please expand the background explanation of the FM model architecture. For AR, you clearly model the parameter size and architecture, but for FM you only mention a paper reference and skip connections. So for the average reader, they must navigate to the other paper to get a quick understanding.
•	In Section 4.2, explain why you modify the AR transformer architecture to have 400M parameters instead of using one of the three configurations in the MusicGen paper of: 300M, 1.5B, 3.3B parameters?
•	In Section 4.2, the cited reference of the FM model does not actually mention how it leverages T5 for text within the arch. It also note the transformer model architecture is 330M parameters. So it seems you are comparing a 400M parameter arch that was originally 300M to a 330M parameter model. Can you help us understand why did you not use the same transformer architecture for the FM model as you did for the AR model? Why setup the experiment with this difference? If intentional, please explain why. With the goal and focus on producing reproducible insight, it would be best to keep the architectures the same or at least ablate it.
•	In Section 4.2 Input Representation, please expand with a little more detail on VAE and quantizer settings or at least mention you ablate different latent sampling rates and have an appendix with more details. Explain why you wouldn’t benchmark with a pre-trained open source implementation.
•	For FADTK eval, please mention and site what backbone you use with the implementation. I assume CLAP_music, but it is unclear.
•	Section 4.3 - Citation of CLAP and footnote URL correspond to two different papers. You should cite correct the paper that corresponding open-source model you are using and not a different paper.
•	Section 4.4 – the batch size of 256 is relatively small for diffusion and flow-models, which likely immediately puts it at a disadvantage.
•	Table 2: What are the ground truth CLAP score? These CLAP scores are very high relative to some open source references sets which can saturate the CLAP score.
•	Throughput definition is unclear  - “throughput (samples / sec)”
•	Section 4, training models on 10 second segments is relatively low and much lower then both current open-source models of MusicGen and Stable Audio Open, do you expect your results to change if training on a longer duration? If so, please comment on it.
•	In Section 5.1, Table 2, try to add some more insight as to why the results are the way they are. When I look at Table 2, I find it very interesting that reducing the latent sampling rate improves performances for basically every metric. This makes sense from a diffusion/flow-matching/AR perspective since there are less “tokens” to model, but it also will result in lower VAE/quantizer performance that is not actually captured in your current presentation of the data. So there’s a clear trade-off. This result also likely suggests that as you increase the latent sampling rate, you likely need more training iterations and/or a bigger batch size since you are effectively modeling more information on the generative model and not the representation model (VAE/quantizer).
•	Section 5.1, when you note that FM requires a large number of steps, you likely will want to limit or caveat this to refer to Section 5.4. You don’t have a symmetrical statement for AR models noted that they take # number of forward passes. Also, in section 5.4, FM models are faster most of the time except for the extreme batch size case, so this statement can be misleading.
•	Section 5.2 qualitative analysis and Appendix D on why control hurts fidelity is great! This focuses on more the deeper insight instead of trying to pick a winner and likely will last the test of time longer.
•	Section 5.4, the definition of throughput of “(samples / sec)” is unclear and should be clarified. The current term of “samples” should be clear. Also, it would be beneficial to use a definition of throughput to be “generated seconds / compute seconds” or the inverse so that it’s more independent of the generation length.
•	In Section 5.4, can you note what is the realistic application of running inference with a batch size of 32-256? Is this for mass creation of AI generated music? From a human perspective, I seems like weighted the speed close to one to ten generations would be more beneficial. In this, case KV caching has little impact and FM are heavily preferred w.r.t., speed.
•	In Section 5.4, it would be useful to hypothesize how inference speed would scale w.r.t. generation length. Ten seconds for generation is relatively low for real use cases.

---

> ### Author Response · Authors · 2025-08-06
>
> We thank you for taking the time to review our work and for the thoughtful feedback.
> As there were many points, we attempted to group them to make this comment shorter.
> Still, this reply spans over two comments due to character limitations.
> Please see our comment below.
>
> ### Regarding reframe takeaways and emphasize insights
> We appreciate the constructive feedback and believe it helped in improving this work. Please see the revised submission which takes the narrative focus concerns under consideration. Included highlights:
> - Reframe of conclusions table, revisions for takeaway paragraphs.
> - Sec 5.1: (i) following reviewer 24at’s suggestion we added an experiment focusing on the sensitivity the FM approach demonstrates w.r.t latent frame rate. (ii) Added comparison vs MusicCaps (Appendix F)
> - Sec 5.4: Added runtime analysis for longer segments, updated conclusions.
>
> ### Regarding experimental design and comparisons to open source models / datasets
>
> - **Diff in model architecture.** Indeed for the FM case the model architecture was slightly shifted by adding UNet-like skip connections, following prior works e.g. [MusicLDM](https://arxiv.org/pdf/2308.01546), [MusicControlNet](https://arxiv.org/pdf/2311.07069), [DiffARiff](https://arxiv.org/pdf/2406.08384), etc. In preliminary experiments we noticed that these skip-connections had a notable impact over model performance, hence for a fair comparison we follow this architectural choice. Regarding the model sizes, we use audiocraft (MusicGen & JASCO’s codebase) as our codebase for the performed experiments. We use the official [“MusicGen-small” configuration](https://github.com/facebookresearch/audiocraft/blob/main/config/model/lm/model_scale/small.yaml) (which has ~420M params total, including the conditioning modules). As the number of additional parameters added due to the u-net like skip connections (+ ~7M params) or the input projection for the temporal conditioning setup (+ ~1M params) are fairly small, we chose to omit such details in order to make it simpler for the reader. We add these details to the appendix.
>
> - **Choice of 32KHz mono setup & stereo modeling.** Our training and evaluation datasets were collected at 32KHz Mono. We chose to follow similar standards as in previous music generation works ([MusicGen](https://arxiv.org/abs/2306.05284), [JASCO](https://arxiv.org/pdf/2406.10970)) who also use Mono 32KHz data. We believe that our observations shouldn’t change much at higher sampling rates. Though not observed in this study, we believe our conclusions would hold for stereo music generation. The vanilla stereo generation application would be to add additional “stream”, and to have either an interleaved modeling like the one presented in MusicGen for the AR case, or a stacked representation (concat L, R latent representations over the channel axis) effectively multiplying the latent dimension by 2. When considering a joint representation of the stereo signal, i.e. the representation model itself is trained to compress and reconstruct a stereo signal, further experimentation would be required to derive conclusions.
>
> - **Evaluation over Musiccaps.** MusicCaps is commonly used but suffers from inconsistent quality due to being sourced from YouTube. Its variability in encoding and loose text alignment makes it unreliable for subtle comparisons. Thus, we chose a high-quality proprietary test set to ensure more meaningful differentiation between models. To support this, we report CLAP similarity and Audiobox-Aesthetics on ground-truth audio-text pairs:
>     |Dataset|CE|PQ|clap|
>     |-|-|-|-|
>     |our eval|7.23±0.57|7.76±0.41|0.34±0.11|
>     |p5|7.05±0.79|7.79±0.48|0.34±0.12|
>     |sttrstk|7.27±0.50|7.84±0.35|0.39±0.09|
>     |musiccaps|6.14±1.44|6.91±1.17|0.28±0.12|
>
>     These show that MusicCaps has higher variance and lower fidelity. PQ correlates with CE (subjective perception), while CLAP highlights audio-text mismatch. Overall, our dataset choices allow for more reliable, fine-grained evaluation of modeling approaches.
>
>     Re-evaluating the models in Table 2 (1M steps, 10 sec, BS 256) over the MusicCaps dataset yielded the following:
>     |FR|Modeling|FAD↓|CLAP↑|PQ↑|PC↑|CE↑|
>     |-|-|-|-|-|-|-|
>     |25|AR|4.10|0.33|7.17|5.13|6.59|
>     ||FM|5.53|0.30|7.42|4.17|6.42|
>     ||FM(VAE)|4.58|0.30|6.99|4.70|6.24|
>     |50|AR|3.55|0.33|7.10|4.93|6.36|
>     ||FM|5.65|0.29|7.39|4.26|6.28|
>     ||FM(VAE)|5.42|0.29|7.10|4.60|6.10|
>     |100|AR|3.52|0.33|6.63|5.27|6.39|
>     ||FM|6.02|0.29|7.47|4.53|6.45|
>     ||FM(VAE)|4.62|0.29|7.00|5.54|6.65|
>
>     While the slight advantage trend AR showed on our eval set, these observations demonstrate a noisy output. Due to the high variance and lower fidelity and text match of the MusicCaps dataset it is hard to determine the significance of the observed performance gaps.

---

> > ### Author Response · Authors · 2025-08-06
> >
> > - **Why don't we compare to open source models.** We agree that comparing our trained models to open-source models could serve as a reference point for our model's performance. We include a 50Hz comparison with MusicGen-small and JASCO open source checkpoints (both have a similar size backbone models). Note 1: MusicGen-small and JASCO-small have the same representation, AR and FM have the same representation; both are (different) EnCodec models. Note 2: JASCO was evaluated with controls dropout (only text).
> >
> >     |Model|FAD↓|CLAP↑|PQ↑|PC↑|CE↑|
> >     |-|-|-|-|-|-|
> >     |MusicGen-small (400M)|0.49|0.34|7.58|5.30|6.98|
> >     |JASCO-chords-drums-melody-400M|0.73|0.39|7.66|5.42|7.00|
> >     |AR 50Hz|0.47|0.40|7.69|5.78|7.24|
> >     |FM 50Hz|0.48|0.40|7.73|5.60|7.20|
> >
> > - **Training with larger batch sizes and longer durations** We followed several prior works that trained text-to-music models with similar batch sizes, e.g. [StableAudioOpen](https://arxiv.org/pdf/2407.14358) and [DiffARiff](https://arxiv.org/pdf/2406.08384) trained a diffusion model with a batch size of 256 and [JASCO](https://arxiv.org/pdf/2406.10970) trained a FM model with a batch size of 336. In addition, we performed preliminary experimentation in which we trained FM models using larger batches; as there were no notable differences at 250K updates we decided to cut down batch sizes to 256 for faster training. See [validation curve](https://bashify.io/i/8bcrrY), and evaluation scores in the table below.
> >     |BS|FAD↓|CLAP↑|PQ↑|PC↑|CE↑|
> >     |-|-|-|-|-|-|
> >     |256|0.47|0.39|7.89|5.54|7.34|
> >     |1024|0.59|0.38|7.74|5.58|7.29|
> >
> >     In sec 5.5 we observe training with longer durations (30s) though we did not extend this experimentation to longer durations, and left such exploration for future work.
> >
> > ### Regarding text clarifications
> > The following comments were included in the paper, we paraphrased some of them in the revised version to improve clarity.
> > - **AR: why delay pattern is needed.** Please see “Training with delay pattern” paragraph in Background section.
> > - **The cited reference of the FM model does not actually mention how it leverages T5 for text within the arch.** technical details for FM are not meant to refer to another paper to describe the architecture but rather cite it as it also used the same methodology; this paragraph also mentions that t5 text embeddings are passed via cross attention. (sec 4.2 backbone model).
> > - **Please expand with a little more detail on VAE and quantizer settings or at least mention you ablate different latent sampling rates and have an appendix with more details** Please refer to Appendix E.3 (in the revised paper), "Latent Representation Models Configurations"
> >
> > ### Other reviewer notes
> > For the following points please see changes in the revised paper.
> >
> > Technical clarifications:
> > - Re-define throughput (generated 10s examples per second)
> > - Elaborate on the connection between diffusion and flow-matching - we have added a short paragraph describing the equivalence of gaussian noise path with stochastic diffusion in high-level and refered to Lipman et al. for the derivation of this property (see Sec 3.3.)
> >
> > Other:
> > - Fix CLAP & fadtk citations and references.
> > - Add citation for “AR models continue to set strong baselines for musical structure and coherence”.
> >
> > We appreciate the constructive critique and we hope these clarifications will address the reviewer’s concerns.

---

### Decision · Action_Editor_sh6b · 2025-08-29

**Recommendation:** Accept as is

**Additional Comments:**

This paper presents a comparative study of autoregressive (AR) and conditional flow matching (CFM) models for generating music from text.
The reviewers raised several concerns, including the need to focus on presenting practical, long-term insights rather than deciding on a "winner"; the validity and transparency of comparisons, such as model architectures; inadequacies in experimental design and reporting; and insufficient discussion of the rationale and impact of tokenizer selection.
The authors addressed these concerns and made the necessary revisions.
These responses and revisions resolved the concerns. Ultimately, the reviewers decided to accept the paper, and the AE agrees with them.

**Audience:**

Yes

**Audience Explanation:**

This research is expected to attract considerable interest from all those working on music generation applications. However, it is unclear what proportion of the TMLR audience is interested in music generation.

**Claims And Evidence:**

Yes

**Claims Explanation:**

This paper presents an empirical study that compares autoregressive (AR) and flow matching (FM) models for generating music from text. Under fair conditions and using identical data and similar architectures, the study quantitatively and qualitatively evaluated multiple aspects, including generation fidelity, adherence to temporal control, interpolation performance, inference speed, and sensitivity to training settings.

The results confirmed that AR models demonstrated superior generation quality, temporal control, and structural stability, while FM models showed advantages in interpolation performance and speed. All claims are supported by control experiments and multifaceted analyses that convincingly illustrate the performance differences and trade-offs between the models.